# The barley leaf rust resistance gene *Rph3* encodes a predicted membrane protein and is induced upon infection by avirulent pathotypes of *Puccinia hordei*

Hoan X. Dinh [1], Davinder Singh[1], Diana Gomez de la Cruz [2], Goetz Hensel [3], Jochen Kumlehn [3], Martin Mascher [3,4], Nils Stein [3,5], Dragan Perovic[6], Michael Ayliffe[7], Matthew J. Moscou [2], Robert F. Park [1✉] & Mohammad Pourkheirandish [8✉]

Leaf rust, caused by *Puccinia hordei*, is an economically significant disease of barley, but only a few major resistance genes to *P. hordei* (*Rph*) have been cloned. In this study, gene *Rph3* was isolated by positional cloning and confirmed by mutational analysis and transgenic complementation. The *Rph3* gene, which originated from wild barley and was first introgressed into cultivated Egyptian germplasm, encodes a unique predicted transmembrane resistance protein that differs from all known plant disease resistance proteins at the amino acid sequence level. Genetic profiles of diverse accessions indicated limited genetic diversity in *Rph3* in domesticated germplasm, and higher diversity in wild barley from the Eastern Mediterranean region. The *Rph3* gene was expressed only in interactions with *Rph3*-avirulent *P. hordei* isolates, a phenomenon also observed for transcription activator-like effector-dependent genes known as executors conferring resistance to *Xanthomonas* spp. Like known transmembrane executors such as *Bs3* and *Xa7*, heterologous expression of *Rph3* in *N. benthamiana* induced a cell death response. The isolation of *Rph3* highlights convergent evolutionary processes in diverse plant-pathogen interaction systems, where similar defence mechanisms evolved independently in monocots and dicots.

[1] The University of Sydney, Faculty of Science, Plant Breeding Institute, Cobbitty, NSW 2570, Australia. [2] The Sainsbury Laboratory, University of East Anglia, Norwich Research Park, Norwich NR4 7UK, UK. [3] Leibniz Institute of Plant Genetics and Crop Plant Research (IPK), Corrensstrasse 3, 06466 Seeland, Germany. [4] German Centre for Integrative Biodiversity Research (iDiv) Halle-Jena-Leipzig, Leipzig, Germany. [5] Center of integrated Breeding Research (CiBreed), Department of Crop Sciences, Georg-August-University, Von Siebold Str. 8, 37075 Göttingen, Germany. [6] Julius Kühn-Institut, Institute for Resistance Research and Stress Tolerance, Erwin-Baur-Strasse 27, 06484 Quedlinburg, Germany. [7] Commonwealth Scientific and Industrial Research Organisation, Black Mountain, ACT 2601, Australia. [8] The University of Melbourne, Faculty of Veterinary and Agriculture, Parkville, VIC 3010, Australia. ✉email: robert.park@sydney.edu.au; mohammad.p@unimelb.edu.au

Global food production is reduced by at least 10% by a wide range of microbial pathogens of plants[1,2]. Deployment of resistance genes has long been considered the most cost-effective and environmentally friendly approach to protect crops against pathogens[1,3,4]. However, the effectiveness of resistance genes is often limited to a few years as pathogens evolve rapidly to acquire virulence that erodes or defeats genetic protection[5]. The constant conflict between host plants and their pathogens shapes genetic diversity in both organisms. Rust pathogens are obligate biotrophic fungi that can grow and reproduce only on living host tissues[6]. They cause devastating losses in agricultural production worldwide[5,7], and remain a major threat to cereal production because of the ongoing evolution of virulence that overcomes genetic resistance and can lead to complete crop loss in extreme epidemic situations[8].

To date, 106 loci conferring resistance to the leaf rust pathogens of wheat (*Puccinia triticina*) and barley (*P. hordei*) have been formally catalogued[5]. Resistance alleles for only 10 of these genes have been cloned with six encoding nucleotide-binding, leucine-rich repeat (NLR) immune receptors[9–14]. The four remaining genes encode an ATP-binding cassette (ABC) transporter[15], a hexose transporter[16], a lectin receptor kinase[17], and a membrane-bound ankyrin repeat protein[18]. At least 28 resistance loci have been catalogued as *Reaction to Puccinia hordei* or *Rph* loci (*Rph1* to *Rph27*)[7,19–21], among which a few, including *Rph3*[7], have been deployed in commercial barley cultivars. Only three (*viz. Rph1*, *Rph15*, and *Rph22*) of these genes have been cloned, in part due to the difficulties imposed by the large and repetitive barley genome, highlighting a knowledge gap in this area. The resistance phenotypes conferred by *Rph* genes range from complete immunity (no visual symptoms) to small uredinia with restricted growth. The *Rph3* locus, previously known as *Pa3*, was first discovered in barley landrace 'Estate' using classical genetics[22]. The locus was mapped on the long arm of chromosome 7H and linked to the morphological $X_a$ locus, the mutant allele confers a Xantha seedling phenotype[23]. Pathotypes with virulence for *Rph3* were detected throughout Europe[24], New Zealand[25], South America, and the Middle East[26]. In Australia, virulence for *Rph3* was first detected in 2009 and has since become common in all barley growing areas[27] (Supplementary Table 1). While *Rph3* provides high levels of resistance to avirulent pathotypes, virulence has arisen independently several times. Nonetheless, it remains a valuable source of resistance that can be deployed in combination with other widely effective resistance genes in regions where virulence is infrequent or absent.

The plant immune system encompasses two layers of defence comprising pathogen-associated molecular pattern-triggered immunity (PTI) and effector-triggered immunity (ETI)[5]. In the current model of the plant immune system, PTI is mediated by receptor-like proteins (RLPs) or receptor-like kinases (RLKs) that are localized on the cell membrane[28,29], whereas ETI is mediated by intracellular sensors such as NLRs that are located in the cytoplasm[30]. In the process of invading mesophyll cells, rust pathogens secrete effector proteins to promote colonization[31]. Some of these effectors are recognized by corresponding receptors encoded by the host. Most of the known intracellular receptors are NLR proteins[32,33] that recognize pathogen effectors by direct[9,10,12,34] or indirect interaction[35–37]. Pathogen recognition is followed by signal transduction through various cascades to activate the immune system and trigger defence response. The vast majority of cloned race-specific resistance genes ("R genes") encode NLRs, and the detailed mechanism of resistance associated with them contains unknown factors[38]. A significant knowledge gap concerns other molecular partners involved in the process of signalling by NLR proteins. Discovering these signalling components could improve the breeding and engineering of

crops for disease resistance. Also, a more comprehensive understanding of the repertoire of plant resistance genes will enhance knowledge of plant-pathogen defence biology and facilitate the diversification of strategies for disease control.

In this study, we isolated the leaf rust resistance gene *Rph3* in barley by positional cloning and mutagenesis. *Rph3* encodes a putative transmembrane protein with no homology at the amino acid level to any plant disease resistance gene isolated to date. We investigated the mechanism underlying this resistance gene and show that *Rph3* is expressed only after a challenge by rust isolates containing the corresponding *AvrRph3* gene. The *Rph3* gene was sufficient to provide resistance to *P. hordei*, and expression of the *Rph3* gene causes cell death in barley and *Nicotiana benthamiana*.

## Results

### *Rph3* is an incompletely dominant gene that confers resistance to *P. hordei*. Barley line BW746 (Bowman*11/Estate) is near-isogenic to cultivar (cv.) Bowman and carries the *Rph3.c* allele from the landrace Estate. Having 10 backcrosses to cv. Bowman, this line theoretically comprises more than 99% of the recipient cultivar genome. Inoculation of seedlings with *P. hordei* pathotypes 5453 P+ (*AvrRph3*), and 200 P− (*AvrRph3*) showed that cv. Bowman is susceptible, and BW746 is resistant to *P. hordei* pathotypes 5453 P+ (*AvrRph3*) (Fig. 1a) and 200 P− (*AvrRph3*). A single introgressed segment from Estate located on chromosome 7H was detected in BW746 by using genotypic data for 19,593 GBS markers[39]. Fungal infection sites observed microscopically at 2-days post-inoculation (dpi) in both, cv. Bowman and BW746 were similar in size and morphology (Fig. 1b, Supplementary Fig. 1a). At four dpi, fungal hyphae were much more abundant in cv. Bowman than in BW746 (Fig. 1b) and differences in fungal biomass accumulation were seen (Fig. 1c) despite macroscopic symptoms being similar between both lines (Fig. 1a vs b). At eight dpi, large colonies were formed in cv. Bowman and most infection sites showed urediniospore production, whereas infection and biomass accumulation were restricted in BW746 and only a few infection sites developed small uredinia (Fig. 1a, b, c). Autofluorescent cell death was not observed in haustorium-infected mesophyll cells of either BW746 or Bowman at either two dpi or eight dpi (Supplementary Fig. 1a). However, infected mesophyll cells of BW746 showed strong uptake of trypan blue stain at four dpi indicating obvious changes in cell membrane permeability that are suggestive of a loss of cell viability in response to fungal challenge (Supplementary Fig. 1b), while infected cells of cv. Bowman showed no change in membrane permeability (Supplementary Fig. 1c). The restricted pathogen development on cv. BW746 was also associated with chlorotic halos around the infection sites and failure to form large uredinia. F$_1$ plants of Bowman crossed with BW746 exhibited an intermediate response to those of BW746 (resistant) and Bowman (susceptible) (Supplementary Fig. 2), suggesting incomplete dominance in the expression of *Rph3*.

### Map-based cloning of *Rph3*. A population of 182 recombinant inbred lines (RILs) from the cross between cv. Scarlett (*Rph3*) and cv. Tallon (*rph3*) was used to investigate the chromosome region encompassing *Rph3* (Supplementary Data 1). The entire population was genotyped with markers previously reported as near the *Rph3* locus in chromosome arm 7HL[40]. Tunable Genotyping-by-Sequencing (tGBS) was used on 42 representative RILs from both resistant and susceptible phenotypic classes (21 lines for each) carrying recombinant chromosomes in the vicinity of the *Rph3* gene, resistant and susceptible bulks each from 10 lines, and the parents. This identified 24 markers linked closely to the *Rph3* locus and delimited it in a physical window of 4.7 Mb based on the reference cv. Morex genome. Several annotated high-confidence genes evenly distributed within this window were

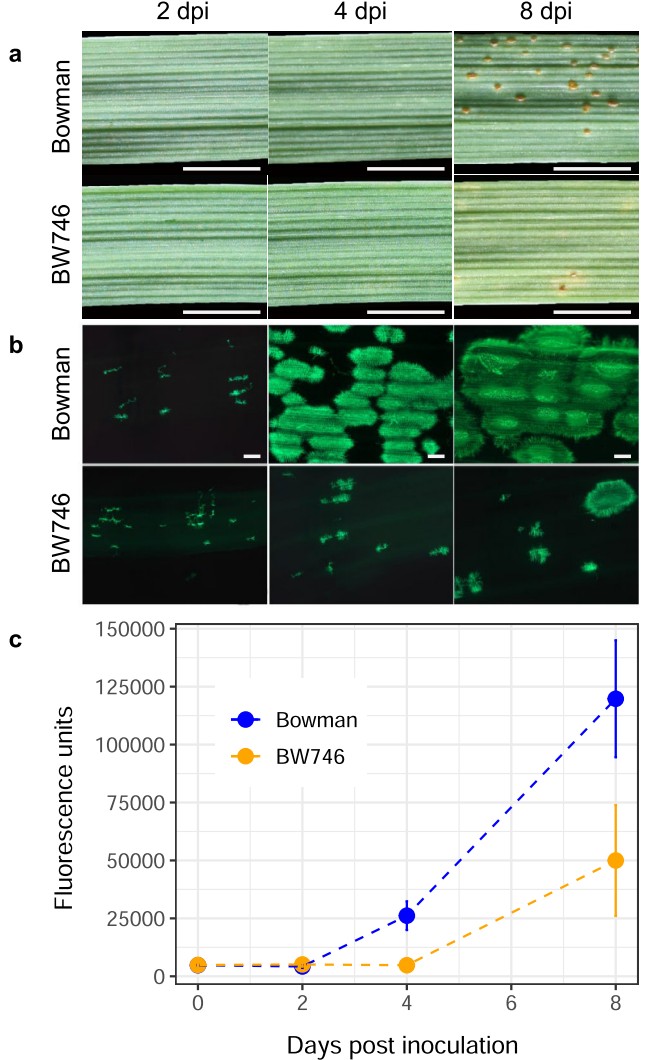

**Fig. 1 Development of *Puccinia hordei* in leaves of cv.** Bowman (*rph3*) compared to its near-isogenic line BW746 (*Rph3*). **a** Segments of infected leaves of cv. Bowman (top) and BW746 (bottom) at 2, 4, and 8 dpi. The scale bar is 0.5 cm. The photos were taken from the same leaves throughout the timepoints. The experiment was repeated five times with similar results. **b** Microscopic visualization of WGA-FITC-stained fungal colonization of mesophyll cells of cv. Bowman and BW746 leaves at 2, 4, and 8 dpi. The scale bar is 200 μm. **c** Quantification of *P. hordei* growth in cv. Bowman and BW746 leaves by the wheat germ agglutinin chitin assay. Fluorescence values for cv. Bowman are shown as blue dots; those from BW746 shown as orange dots. Values represent means ± SD (shown as the error bars) (*n* = 4).

selected to design markers to enrich the genetic map of *Rph3*. After screening 10,411 F$_2$ individuals from six populations (Supplementary Tables 2, 3) with flanking markers MLOC_005 and MLOC_040, 367 recombination events were identified. Phenotyping of the recombinant families delimited the *Rph3* locus to a 0.22-cM interval flanked by markers MLOC_004 and MLOC_023 with 45 recombination events between them (Supplementary Table 3). Nine additional markers developed in this region (Supplementary Data 2) based on the reference genome mapped the *Rph3* locus to a 0.02-cM interval between markers MLOC_190 and MLOC_389 with two and three recombinants to the *Rph3* locus, respectively (Fig. 2, Supplementary Data 3). The three recombinants between MLOC_389 and *Rph3* were

confirmed by sequencing (Supplementary Fig. 3). The physical delimitation of the *Rph3* gene was carried out in cv. Barke that has been shown to carry the resistance gene based on multi-pathotype tests in this study (Supplementary Table 2) and the availability of draft genome sequence[41]. The *Rph3* locus was located in a physical window of 8,519 bp based on the cv. Barke (*Rph3*) genome sequence[41]. This window of 8.5 kb was rese-quenced in all resistant parents from six *Rph3* mapping popula-tions and cv. Barke by the Sanger procedure and demonstrated an identical 8.5 kb sequence without any polymorphisms among all seven resistant lines. There was no annotated gene within the region based on the reference genome annotation for cv. Morex (v2.0 2019)[42]. Manual de novo annotation of the 8.5 kb interval of cv. Barke using FGENESH software identified two open reading frames designated as *ORF1* and *ORF2* (Fig. 2), predicted to encode proteins with 101 and 276 amino acids, respectively.

**Forward genetic screen for loss-of-function of *Rph3*-mediated resistance.** To determine if *ORF1* and/or *ORF2* were required for the resistance, two ethyl methane sulphonate (EMS) mutagenized populations were produced using two resistant lines, BW746 and cv. Henley (*Rph3*). Six altered phenotype mutant families were identified among 850 M$_2$ spikes screened with *Rph3*-avirulent pathotype 5453 P- (Supplementary Table 4). Resequencing of the 8.5 kb *Rph3* region, including *ORF1* and *ORF2* in the six homo-zygous mutants, revealed four lines with a single nucleotide change in *ORF2*. The two remaining lines without changes within the locus were not allelic with the four known mutants (Sup-plementary Table 5). Phenotypic screening of M$_3$ populations of the four mutants with altered sequences in *ORF2* with an *Rph3*-avirulent pathotype confirmed that M198 and M466 were fully susceptible, while M167 and M181 displayed intermediate responses (Fig. 2). Mutant line M198 encoded a truncated protein due to the formation of a new stop codon at position 72, and line M466 had a nucleotide change at the fifth nucleotide in the first intron after the splicing junction. The mutant line M466 did not survive, and the changes in the protein structure of the RPH3 protein could not be examined. Of the other two mutants, M181 had an L93 > F amino acid substitution and M167 had a P126 > L substitution. Uredinia formed by the *Rph3*-avirulent pathotype on plants homozygous for each of these latter mutants were significantly larger than those formed on the resistant parents (Supplementary Fig. 4). These mutant phenotypes were con-sistent with changes at the molecular level: alterations in protein structure involving an early stop codon (M198) and a predicted splicing variant (M466) resulted in fully susceptible responses. In contrast, the single amino acid substitutions (M167 and M181) resulted in intermediate responses. All these independent point mutations occurred in *ORF2*, and no change was detected in *ORF1* or the intergenic region (8.5 kb physical window) in any of the six altered phenotype mutants. These results demonstrated that *ORF2* is required for *Rph3*-mediated resistance.

**Transgenic complementation of *Rph3*.** To determine if *ORF2* is sufficient to complement the lack of *Rph3* for resistance to *P. hordei*, we conducted a complementation test using the complete genomic coding sequence of *ORF2* driven by its native promoter. Splice alignment of RNA-seq revealed that *ORF2* consisted of an 831 bp coding sequence and 254 bp 5'-, 292 bp 3'- untranslated regions (UTRs). A 7196 bp DNA fragment containing the entire transcribed region of *ORF2* including the native promoter (3146 bp upstream region) of the resistant cv. Barke (Supple-mentary Fig. 5) was transformed into the susceptible barley cv. Golden Promise (*rph3*). The T-DNA construct was unambigu-ously detected based on PCR amplification of a selectable marker

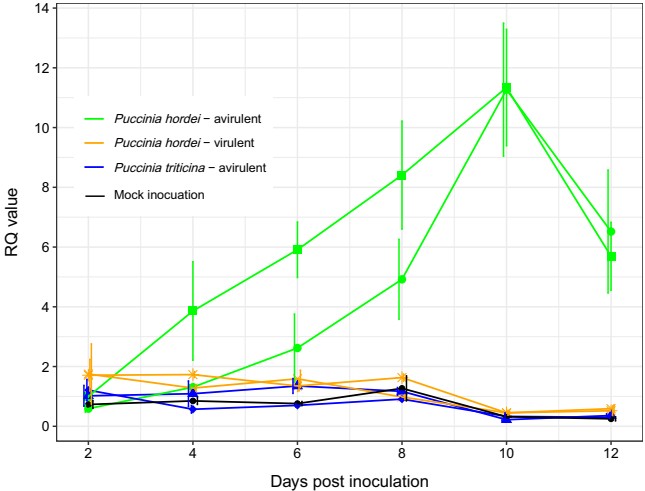

**Fig. 2 Map-based cloning of barley leaf rust resistance gene *Rph3*.** A high-resolution genetic linkage and physical map of the *Rph3* locus were constructed based on segregation among 10,411 F₂ individuals. Forty-five recombinants were found between flanking markers MLOC_004 and MLOC_023. The *Rph3* gene was physically located in an 8519 bp interval based on cv. Barke reference sequence. Two putative genes identified within the window are shown as *ORF1* and *ORF2*. Four independent EMS-induced mutants within the coding sequence of *ORF2* indicated that the *ORF2* was required for *Rph3* resistance.

gene in 16 primary (T₀) transgenic from a total of 20 plants generated under selective conditions (Supplementary Table 6; Supplementary Fig. 6). The presence of the transgene in the T₁ generation co-segregated with a resistant response to the *Rph3*-avirulent pathotype based on a specific marker detecting the *Rph3* resistance allele. This genetic transformation experiment demonstrated that *ORF2* complemented the lack of *Rph3* in cv. Golden Promise. Taken together, high-resolution and physical delimitation, four independent mutants, and transgenic complementation results provide convergent evidence that *ORF2* was *Rph3*.

**Rph3 is induced by *P. hordei* isolates avirulent for *Rph3*.** The *Rph3* transcript was not found in any published barley RNA-seq, full-length cDNA, or expressed sequence tag (EST) database. Transcript of *Rph3* was detected in leaves of the resistant line BW746 inoculated with *P. hordei* pathotypes avirulent for *Rph3* by RT-qPCR (Fig. 3). In contrast, no transcript was detected in leaves inoculated with either *Rph3*-virulent pathotypes or in mock inoculations (Supplementary Fig. 7), which implies *Rph3* is only induced during an incompatible interaction. *Rph3* transcripts were detected in plants of BW746 when challenged with two different *Rph3*-avirulent pathotypes (200 P− and 5453 P+), but not when inoculated with two different *Rph3*-virulent pathotypes (5457 P+ and 5656 P+). Transcripts were also not detected in inoculations with the wheat leaf rust pathogen *P. triticina* (pathotypes 26-0 and 104-1,2,3,(6),(7),11,13). These results demonstrate that expression of *Rph3* is induced explicitly by infection with an *Rph3*-avirulent *P. hordei* pathotype (Fig. 3). Moreover, *Rph3* expression was detected only in infected tissue, indicating that a signal could not be transmitted to non-infected parts of the same plant (Supplementary Fig. 8). Examination of transgenic families showed that the expression profile of the *Rph3* transgene in the T₁ segregating progenies was well aligned with the presence/absence of the transgene detected by the diagnostic *Rph3* marker (Supplementary Fig. 9). The exogenic *Rph3* expression level in transgenic plants was much stronger than that of the *Rph3* allele in cv. Estate when challenged with an *Rph3*-avirulent pathotype, which could be due to the copy number of the transgene or to a transcript enhancing element in the vicinity of the inserted gene (Supplementary Fig. 9). Expression was not detected for any *Rph3* homolog in the susceptible haplotype (cv. Morex) during infection regardless of the rust pathogen used (Supplementary Fig. 10). Similarly, transcripts of the putative *ORF1* were not detected in any treatment. Taken together, these experiments showed that *Rph3* is expressed explicitly in barley genotypes carrying the *Rph3* resistance allele, when challenged with an *Rph3*-avirulent *P. hordei* pathotype and that upregulation of the gene occurs exclusively in infected tissue.

**Fig. 3 Transcript levels of *Rph3* detected by RT-qPCR during 2–12 dpi in response to virulent and avirulent pathotypes.** The *Rph3* gene was upregulated when the leaf was infected by *Rph3*-avirulent *P. hordei* pathotypes (green dot = 200 P−, green square = 5453 P+), whereas the transcript levels were unchanged when the leaf was infected by *Rph3*-virulent pathotypes (yellow cross = 5656 P+, yellow asterisk = 5457 P+) and *P. triticina* (blue diamond = 26-0, blue triangle = 104-1,2,3,(6),(7),11,13). The transcript levels of *Rph3* in un-inoculated seedlings (mock inoculation) is shown in the black line. Values represent means ± SE (shown as the error bars) (*n* = 3). Samples inoculated with pathotype 5453 P+ at two dpi were used as calibrations to calculate the relative quantification (RQ) values using the delta-delta method with RQ = $2^{-\Delta\Delta Cq}$. The ADP-ribosylation factor gene was used as a normalizer.

**Bioinformatic and phylogenetic analysis of the *Rph3* gene family.** BLAST searches of RPH3 amino acid sequences against the National Center for Biotechnology Information revealed no matches to the Conserved Domain Database (CDD v3.18—55570 PSSMs) using the default expected (E)-value. This suggests that the RPH3 protein is highly divergent among different plant species, lineage-specific, or not annotated due to a lack of molecular evidence such as RNA-seq. RPH3 secondary structure predictions from three independent programs (TMHMM, TMPRED, and Protter) suggested an insoluble protein comprising 5–7 transmembrane helices (Supplementary Figs. 11, 12), indicating that RPH3 is likely an integral membrane protein. The analysis using HHpred revealed 14 hits from bacteria, yeast, fungi, mungbean, fish, house mouse, and humans, with the probabilities ranging from 20.1 to 54.8% (Supplementary Table 7). Most of these were consistent with the *Rph3* gene encoding a transmembrane protein, in agreement with the prediction made by the

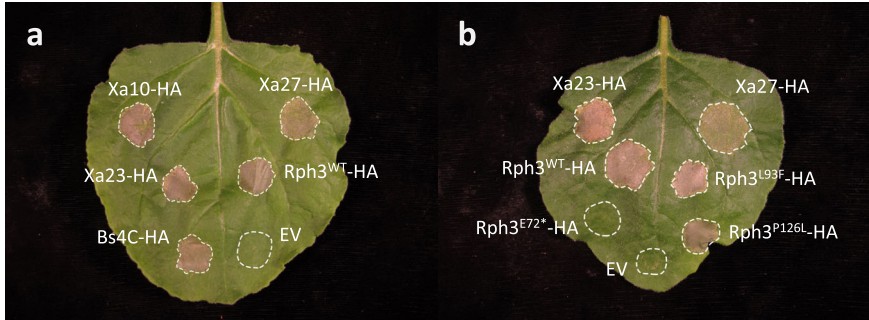

**Fig. 4 Rph3 induces cell death in N. benthamiana. a** Transient expression of executor resistance genes (*Xa10*, *Xa27*, *Xa23*, and *Bs4C*) and *Rph3* (C-terminal HA-tagged) induce cell death in *N. benthamiana* under a mas promoter. **b** Non-synonymous *Rph3* mutants M167 (L93F) and M181 (P126L) retained the ability to cause cell death in *N. benthamiana*, whereas the truncation mutant M198 (E72*) did not. *Xa23* and *Xa27* were used as controls for induction of cell death. The experiment was performed three times and included infiltration of two leaves of three to four plants with similar results.

TMHMM, TMPRED, and Protter tools. The RPH3 protein shares 14% similarity with a nucleoside transport protein (CNT$_{NW}$) in the bacterium *Neisseria wadsworthii*[43], 13% identity with the citrate transport protein (SeCitS) in the bacterium *Salmonella enterica*[44], and 18% identity with the electron transport protein (VrIII$_2$) in mungbean (*Vigna radiata* var. *radiata*)[45], implying that the RPH3 protein may be involved in the transportation process via the membrane.

A BLASTX search against the non-redundant database using the cDNA of *Rph3* as a query returned seven hits with different levels of identity. The 9-cis-epoxycarotenoid dioxygenase (HORVU_NCED) protein from barley shares 46% identity with RPH3. Two sequences with similarity to RPH3 were retrieved from *Aegilops tauschii*, consisting of LOC109787323 and LOC109787282, and one from *Brachypodium distachyon*, Bradi1g31183.3. The analysis using a hidden Markov model HHMER with the RPH3 amino acid sequence as query showed seven hits that are consistent with the results from the BLASTX search. No ortholog was identified in *Brachypodium stacei*, suggesting that this gene family experiences gene loss in independent lineages. BLASTN against the reference genome of various crop species revealed homologs of *Rph3* in each of the three genomes A, B, and D of bread wheat (*Triticum aestivum*), and one homolog in oat (*Avena sativa*). BLASTN against the barley Morex v2.0 reference genome[42] found four putative homologous genes of *Rph3*, all located within 98.8 kb flanked by markers MLOC_190 and MLOC_389 in chromosome arm 7HL. These four genes *HORVU_ORF5*, *HORVU_ORF10*, *HORVU_ORF11*, and *HORVU_ORF12*, and their sequence similarities are described in Supplementary Fig. 13 figure legend.

The phylogenetic relationship between the RPH3 protein and the four cereal homologs suggests that the RPH3 protein evolved the ability to confer resistance against *P. hordei* within barley after the divergence of wheat and barley (Supplementary Fig. 13a). However, putative orthologues could be involved in disease resistance in the related species. Analysis of motif composition of RPH3 and its homologs/paralogs using the Surveyed conserved motif ALignment diagram and the Associating Dendrogram (SALAD) showed eight conserved motifs (Supplementary Fig. 13a), and seven transmembrane helices overlapped all of these motifs except for motifs 5 and 8. Among all, motifs 1–3 were present in almost all related proteins, of which motif 1 has two N-myristoylation sites, one phosphorylation site of protein kinase C, and two phosphorylation sites of casein kinase II (Supplementary Fig. 13b). Although considerably larger than RPH3 (276 aa), the wheat homolog TraesCS7D_RPH3_LIKE (401 aa) located on chromosome 7D shares all motifs with RPH3 and in the same order, suggesting they are orthologs.

Grass species diverged from a common ancestor about 60 million years ago[46] and have considerable variation in chromosome number, genome size, and sequence. However, most of the genes present in grass species are conserved, and the gene order among them is mainly collinear[47,48]. The long arm of barley chromosome 7H that harbors *Rph3* is syntenic with the long arm of chromosome 7 in the wheat A, B, and D genomes[49]. We showed that micro-synteny is well conserved in the vicinity of *Rph3* between barley and wheat genomes (Supplementary Fig. 14). Orthologs of the *Rph3* gene were found in the wheat A, B, and D genomes within the expected locus, of which the copy from the D genome has motifs in identical order to RPH3, and the two proteins share 88% similarity at the amino acid level (Supplementary Fig. 13a). Four loci conferring resistance to *P. triticina* (causal agent of wheat leaf rust) on one or other long arm of wheat chromosome 7 have been designated, namely *Lr14a-b* (7BL), *Lr19* (7DL), *Lr20* (7AL), and *Lr68* (7BL)[50–53]. Among these loci, *Lr68* confers adult stage resistance while the other three loci confer all-stage resistance. None of these genes is located in a region homologous to the *Rph3* gene (Supplementary Fig. 14). This suggests that either *Rph3* gained a role in immunity post divergence, or alternatively, insufficient sampling has been performed in Triticeae species to identify functional orthologs.

**Rph3 induces cell death in N. benthamiana.** The expression of *Rph3* in the presence of the corresponding avirulence gene but apparent lack of expression when avirulence is lacking is reminiscent of executor gene resistance to *Xanthomonas* spp. conferred by genes such as *Bs3*[54], *Xa10*[55], *Xa23*[56], *Xa27*[57], and *Bs4C*[58]. We performed heterologous expression of *Rph3* in *N. benthamiana* and found that it caused cell death when transiently expressed under the *MasΩ* promoter (Fig. 4). This cell death phenotype was comparable to cell death in *N. benthamiana* induced by overexpression of *Xa10*[55], *Xa23*[56], and *Bs4C*[58]. Previously, heterologous expression of *Xa27* was not shown to cause cell death in *N. benthamiana*[57]. We found that this absence of cell death is likely dependent on expression level, as the *MasΩ* promoter was sufficient for *Xa27*-mediated cell death in *N. benthamiana* (Fig. 4a). Expression of *rph3* alleles identified from the loss-of-function mutagenesis screen indicated that early truncation mutant M198 (E72*) did not cause cell death, whereas the non-synonymous mutants M167 (L93F) and M181 (P126L) caused cell death in *N. benthamiana* (Fig. 4b, Supplementary Fig. 15). This result matched quantitative phenotyping results of the mutants, where mutant M198 (E72*) has the most significant effect on response in showing complete susceptibility, whereas mutants L93F and P126L are partial loss-of-function with a reduced level of resistance.

**Transcription dynamics of *Rph3*-mediated resistance at two days post-inoculation**. We performed RNA-seq analysis of cv. Bowman and BW746 to measure the response of barley to *P. hordei* in the presence and absence of *Rph3* two days after inoculation with *Rph3*-avirulent *P. hordei* pathotype 5453 P+ or the application of oil (mock). Differentially expressed genes were identified for every pairwise comparison of genotype and treatment using a false discovery rate of 5% (Supplementary Fig. 16). RNA-seq reads for *Rph3* were detected in two of three replicates of BW746 inoculated with *P. hordei* and not seen in any other treatment. More genes were differentially expressed in the incompatible interaction among treatments than in the compatible interaction at 2-days post-inoculation. This comparison also produced the most significant number of unique differentially expressed genes among treatment comparisons (3004 DEG). Gene ontology enrichment analysis found that upregulated genes are associated with several biological processes related to transport, such as vesicle-mediated ($p_{adj} = 8.4e$-25) and protein transport ($p_{adj} = 5.2e$-15). In contrast, enrichment in downregulated genes was localized to the plastid ($p_{adj} = 7.5e$-36) and associated with photosynthesis ($p_{adj} = 2.1e$-4) (Supplementary Data 4). This indicates that *Rph3*-mediated resistance is correlated with upregulation of endomembrane trafficking components, which might contribute to the immune response. However, the actual genes associated with pathways, or the mechanism of *Rph3*-mediated resistance could not be determined due to a large number of DEG.

**Allelic variation in *Rph3***. The *Rph3* region located between MLOC_190 and MLOC_389 in the susceptible barley cv. Morex encompasses a physical interval of 98,478 bp compared to 8519 bp in the resistant cv. Barke (Supplementary Fig. 17a, b). Four homologous sequences of the *Rph3* gene were found in cv. Morex (Supplementary Fig. 17c). All four *Rph3* homologs encode proteins of unknown function. Analysis of the promoter region of *Rph3* in cv. Barke (*Rph3* allele) used for transformation compared to cv. Morex (*rph3* allele; HORVU.MOREX.r3.7HG0741050) revealed 81.7% (2598/3179) sequence identity. Mapping of putative transcription factor binding sites identified substantial variation in putative sites between the Barke and Morex haplotypes (Supplementary Fig. 18). The BLASTN of the *Rph3* gene against the whole genome of cv. Barke revealed only one hit in the barke_contig_512435, the *Rph3* gene. A PCR primer pair based on the draft reference genome sequence of cv. Barke[59] was designed to amplify the complete coding and intron sequences of *Rph3* (Supplementary Data 5). These primers were applied to a collection of 78 barley accessions comprising 41 lines with and 37 without *Rph3* and representing all known *Rph3* alleles, including *Rph3.c*, *Rph3.aa*, and *Rph3.w*[60]. All 41 lines postulated to carry *Rph3* genes were PCR-positive, and all 37 lines postulated without *Rph3* were PCR-negative for the designed primers (Supplementary Data 6). Resequencing of 41 PCR-positive showed that the entire DNA sequence was identical among all resistant accessions. This finding suggests a monophyletic origin of *Rph3* within cultivated barley. The responses of three Bowman NILs carrying three different postulated alleles of *Rph3* to pathotype 5453 P+ (avirulent for *Rph3*) of *P. hordei* were the same (Supplementary Fig. 19). Therefore, we conclude that all these stocks originated from the same ancestor and transcribed one unique isoform of *Rph3*.

Analysis of GBS markers using a worldwide barley collection of 20,607 accessions identified a single paired GBS marker landing on the *Rph3* gene. This paired GBS marker (gRph3_I1E2 and gRph3_E2I2; Supplementary Table 8) was detected in 134 accessions comprising 32 landraces, 70 cultivars, 14 breeding lines, 15 wild accessions, 1 semi-wild accession, and two other genotypes (Supplementary Data 7). The landraces and breeding lines with *Rph3* were from many parts of the world, but the cultivars were mostly from Europe (especially Germany with 27 accessions). The wild accessions were collected in Israel (9 accessions), Syria (8 accessions), Jordan (2 accessions), Greece (2 accessions), or had unknown origins (4 accessions)[61]. Haplotypes identified with this approach had an identical sequence for the GBS markers. This GBS marker was applied to the 314 Wild Barley Diversity Collection (WBDC) population and identified 10 accessions carrying *Rph3* (Supplementary Data 8)[62]. Simultaneously, a dominant marker (MLOC_400, Supplementary Data 2) based on the *Rph3* gene sequence confirmed the presence of the dominant *Rph3* allele in all 10 accessions carrying the GBS marker and identified five additional accessions (WBDC044, WBDC094, WBDC238, WBDC254, and WBDC260) (Supplementary Data 8). Sequence alignment of GBS markers for the five WBDC accessions not previously identified in the k-mer analysis found eight to nine SNPs relative to *Rph3*. Three additional haplotypes were identified as Hap2: WBDC094 and WBDC254; Hap3: WBDC238 and WBDC260; and Hap4: WBDC044. Even though the diagnostic marker MLOC_400 is positive, attempts to amplify the full *Rph3* alleles in WBDC044 and WBDC094 were unsuccessful (Supplementary Fig. 20) probably due to nucleotide variation at the primer binding site. Whole-genome sequencing in future studies will be required to dissect the locus further. The identification of multiple sequence variations within wild barley suggests that additional allelic variants of *Rph3* may exist. These findings also indicated that the *Rph3* gene likely originated from wild barley in Israel, Syria, Jordan, or Greece, from which it was introgressed into cultivated barley germplasm.

## Discussion

Here, we have identified the gene underlying *Rph3*-mediated resistance to *P. hordei* using map-based cloning, mutagenesis, and transgenic complementation. This gene is exclusively expressed when the plant is attacked by avirulent pathotypes, and expression of *Rph3* triggers cell death in barley and *N. benthamiana*. The *Rph3* gene encodes a small protein of 276 amino acids with multiple predicted transmembrane helices and contains no conserved domains of any resistance gene protein families known to date. The expression profile and structural characteristics of the encoded RPH3 protein are reminiscent of TALE-activated executor resistance genes. However, the executor genes that have been reported to date are involved in resistance to bacterial diseases in rice and pepper.

Most cloned disease resistance genes are expressed constitutively[63,64]. Constitutive expression was observed in genes conferring resistance to various pathogens, including bacteria[65–67] and fungi[68–71]. However, the expression of some resistance genes is induced by an external factor, and such genes can be divided into two subgroups. The first subgroup consists of genes whose expression is induced by an avirulent pathotype, a virulent pathotype, or physical damage. Two examples of this are *Xa1*, which confers resistance to *Xanthomonas oryzae* in rice, and *Ve1*, which confers resistance to *Verticillium dahlia* strain Vd1 in tomato. These two genes are induced upon pathogen infection irrespective of pathogenicity, as well as by physical damage[68,72]. The second subgroup consists of genes whose expression is induced exclusively in the presence of avirulent pathogen strains or pathotypes. This phenomenon has been reported for genes conferring resistance to plant viruses, bacteria, and fungi, and the barley gene *Rph3* belongs in this subgroup. A particular induction of a resistance gene to an avirulent pathogen pathotype in the second subgroup has been observed in

only a few systems[56,73]. Expression of the *N* resistance gene in tobacco was induced by TMV infection but not by Potato Virus Y[74]. Induction by an avirulent pathotype only has been documented for genes conferring resistance to fungal pathogens, including the barley mildew pathogen *Blumeria graminis* f. sp. *hordei*[75], the sunflower downy mildew pathogen *Plasmopara halstedii*[76], the rice blast pathogen *Magnaporthe oryzae*[77], and the wheat leaf rust pathogen *Puccinia triticina*[18]. The most comprehensively characterized class of resistance genes in this group are the rice genes *Xa7, Xa10, Xa23*, and *Xa27*, conferring resistance to the bacterial pathogen *X. oryzae* pv. *oryzae*, and the pepper genes *Bs3* and *Bs4C* conferring resistance to *X. campestris* pv. *vesicatoria*. These *Xanthomonas* resistance genes are activated by corresponding transcriptional activator-like effectors (TALE) secreted by avirulent strains[54,56,57,78]. TALE-activated resistance genes were designated "executor" genes as they are solely involved in triggering a plant immune response[79]. In this study, the *Rph3* gene is expressed only upon infection with a pathotype of *P. hordei* carrying the matching avirulence gene. Like *Rph3*, all currently cloned executor genes encode transmembrane proteins. The similarity in both expression profile and transmembrane domains suggests a similar resistance mechanism. We hypothesize that *Rph3*-avirulent *P. hordei* pathotypes produce an effector, *AvrRph3*, that directly or indirectly triggers the expression of the *Rph3* gene. Further work is required to demonstrate this, in particular, the isolation of *AvrRph3*. It will be critical to determine whether AvrRph3 has the capacity to bind DNA and specifically interact with the promoter of *Rph3* or alternatively if AvrRph3 induces *Rph3* expression through earlier transcriptional components such as transcription factors, the Mediator complex, or RNA polymerase II.

Effectors secreted by pathogens target host proteins to enhance infection. On the other hand, plants evolved resistance genes with promoter sequences that are targeted by effector proteins to initiate defence responses, including cell death. This co-evolutionary process has led to host decoy genes, the proteins which mimic an operative effector target to intercept the pathogen effector[80]. In plant-*Xanthomonas* spp. interactions, genes encoding executor proteins facilitate an immune response including promoter sequences similar to host virulence targets. In this context, the promoter of the executor acts as a decoy to the original host target[80]. This model suggests that executors only function when pathogen effectors are present, do not contribute to pathogen fitness in the absence of the cognate R protein, and potentially have an exclusive role in plant immunity[80]. Among the executor genes mentioned above, the *Bs3* gene was suggested to function as a decoy. To date, no other function has been associated with the pepper *Bs3* gene other than resistance to *Xanthomonas*. Inactivity of the *Bs3* gene in the absence of the *AvrBs3* effector supports its exclusive biological function[54]. AvrBs3 targets several promoters, including the promoter of gene *Upa20*[81]. AvrBs3-mediated expression of *Upa20* leads to misregulation of cell size in pepper (hypertrophy)[81]. Notably, both *Bs3* and *Upa20* have the same promoter element, an *upa*-box (TATATAAACCN$_{2-3}$CC), which is targeted by the AvrBs3 effector[54]. In this case, the promoter of the *Bs3* gene acts as a decoy that mimics the target of *AvrBs3* (promoter of *Upa20*), and based on that, traps this effector and activates transcription to trigger the defence response. The *Rph3* gene may not have any function in the absence of an *AvrRph3* effector as we could not detect its expression among publicly available barley RNA-seq databases. Although neither the AvrRPH3 protein nor the operative target of this protein was identified, the similar expression pattern between *Rph3* and *Bs3* indicates that they may work similarly. Further work is required to determine if *Rph3* is

expressed in a unique developmental context or whether it has an exclusive role in plant immunity.

Plants have evolved proteins that recognize pathogen attacks and trigger immune response pathways to defend against invaders upon pathogen detection[82]. Of the 20 resistance genes isolated from wheat and barley that confer race-specific rust resistance, 17 encode NLR proteins[5,83,84], one of the largest and most diversified plant disease resistance gene families[85,86]. The exceptions are stem rust resistance genes *Rpg1* from barley[87] and *Sr60* from diploid wheat, both of which encode proteins with two protein kinase domains in tandem[83], and leaf rust resistance gene *Lr14a* from hexaploid wheat that encodes a protein with 12 ankyrin repeats[18]. The *Rph3* gene is a new class of resistance genes that shows no similarity to any of these genes. The RPH3 protein is predicted to contain five to seven transmembrane helices depending on the prediction tools. The genes *Lr34/Yr18/Sr57/Pm38* and *Lr67/Yr46/Sr55/Pm46* conferring rust and mildew resistance in wheat[15,16], and *Xa7, Xa10, Xa23, Xa27*, and *Bs4* (executor genes) conferring resistance to *Xanthomonas* sp. in rice and pepper[73,88], also encode proteins with multiple transmembrane helices. While *Lr34/Yr18/Sr57/Pm38* and *Lr67/Yr46/Sr55/Pm46* are race-non-specific, executor genes are race-specific. Cloned executor genes encode small proteins (113–342 aa) that are predicted to contain transmembrane helices[55–57,78,88]. The Bs3 protein shows a high level of similarity to flavin monooxygenases[79], whereas the other executor proteins showed no significant sequence homology to any known resistance protein[56,88]. Our study demonstrated that the RPH3 protein appears to cause cell death in barley and in the heterologous system *N. benthamiana*. Cell death can be directly prompted by the RPH3 protein or indirectly via triggering a defence pathway. Previous work has shown that executor genes (*Xa7, Xa10, Xa23, Bs3*, and *Bs4C*) trigger cell death in both their host (rice or pepper) and in *N. benthamiana*[54–56,78,88]. Although *Xa27* was reported to trigger cell death only in rice[57], we found that it does trigger cell death in *N. benthamiana* when driven by the *MasΩ* promoter, suggesting that the expression level is essential for efficacy. XA27 was found in the apoplast, whereas other executor proteins were localized in the endoplasmic reticulum[55,56,78,79,89]. Furthermore, executor genes trigger programmed cell death in different ways; for example, Bs3 causes cell death via the accumulation of salicylic acid and pipecolic acid[79], whereas cell death attributed to XA10 and XA23 is related to cellular $Ca^{2+}$ homeostasis. The mechanisms underlying cell death mediated by BS4C, XA27, and RPH3 remain unknown. The typical features of RPH3 and known executors, including similar expression patterns, small proteins with predicted transmembrane helices, and cell death induction, suggest that the RPH3 protein could be an executor.

Major bottlenecks in genetic variation in many crops were caused by domestication and many variations remained in the wild gene pool[90,91]. Wild barley, which crosses freely with cultivated barley, is a well-known source of allelic variation[91,92]. The *Rph3* gene is a functional allele that confers resistance, and its semi-dominant behavior can be accounted for as a gene dosage effect. Resistance and susceptibility alleles could result from a point mutation (loss of function or gain of function), gene duplication followed by neofunctionalization (gain of function), or be of independent origin (unequal recombination, insertion, deletion, or inversion). The significant differences in the structure between the resistant (*Rph3*) and susceptible (*rph3*) alleles at the DNA level (8.5 vs 98.5 kb) plus many nucleotide substitutions within the causal gene (37 SNPs between the coding sequence of *Rph3* and its most similar gene *ORF10*) imply ancient, independent origins. The *Rph3* resistance allele was detected based on sequence analysis in wild barley accessions collected from the

Eastern Mediterranean and Greece. The gene in modern cultivars originated from two donors, cv. Aim and landrace Estate, both of which are spring type, six-rowed, and came from Egypt[22,93]. The two lines are accessioned as HOR 2470 and HOR 2476 in the barley collection at IPK, and their pedigrees are unknown. The best explanation would be that the gene was introgressed into cultivated barley from wild barley in or around Egypt via hybridization. This hybridization could have been a result of deliberate crossing by a farmer/breeder to introduce a new beneficial allele or random outcrossing between a cultivar and wild relatives growing as a weed in the vicinity followed by deliberate selection by a farmer. It is impossible to separate these hypotheses due to a lack of information about the origin of both accessions. The sequence identity of Rph3 among all 41 resistant lines of cultivated barleys from diverse sources including cv. Aim and landrace Estate indicate a single introgression event. Of interest, while the alleles Rph3.c, Rph3.aa, and Rph3.w were designated based on differing origins[60] all show identical specificity with Australian isolates of P. hordei, and all were found to share 100% sequence identity.

Analysis of variation in the Rph3 allele in wild barleys collected from different geographical areas may allow discovering of other functional alleles of Rph3, allowing direct mining of genetic diversity to discover new resistance alleles to protect barley from P. hordei. Identifying five wild barley accessions carrying polymorphisms in a GBS marker tightly linked to Rph3 suggests that additional alleles of Rph3 may exist in (wild) barley. Further genotypic and phenotypic characterization of genetic diversity is required to determine if these represent novel functional alleles with different specificities or are equivalent to Rph3. The evolution of the Rph3 gene can be further investigated by examining its conservation across species within the Triticeae, and if possible, other Poaceae species to identify the origin of this protein family.

Cloning studies have shown that non-durable resistance genes tend to be NLRs. The current study demonstrates that other types of resistance genes are also vulnerable to evolving pathogens and that much remains to be learned about the durability of resistance genes. Here we hypothesize that the fungus encodes TAL-effector-like proteins, or alternatively, targets other components of the plant transcriptional machinery that precisely activates Rph3 expression. Further studies are required to test these hypotheses. With breakthroughs in gene engineering, the isolation of Rph3 provides an additional resistance gene to include in transgenic cassettes for gene pyramiding. This study suggested that the Rph3 gene has a single origin and was introgressed from wild barley into the cultivated gene pool. Furthermore, engineered executor genes in rice and pepper that contained additional TAL- effector binding sites showed increased resistance specificity. If Rph3 gene induction is also due to the binding of TAL-effector-like proteins to the promoter, a similar strategy could be used to increase the resistance specificity of this protein to other P. hordei pathotypes or other plant diseases.

## Methods

**Histology.** The procedure followed Ayliffe et al. 2011[94] with slight modifications. Segments of 3–4 cm of first leaves from cv. Bowman and BW746 inoculated with the P. hordei pathotype 5453 P+ were harvested at 2, 4, and 8 dpi. The collected leaf samples were autoclaved in 50 ml screw-cap tubes containing 25 ml of 1 M potassium hydroxide (KOH) at 121 °C for one hour to remove chlorophyll. After autoclaving, the KOH solution was gently removed, and the leaf samples were twice gently washed with Tris-HCl (50 mM, pH 7.0) before adding 10 ml of the same Tris buffer to neutralize the samples. Before staining, most of the Tris buffer was removed to leave the tissue at minimum volume. A 1 mg/ml solution of WGA-FITC was added to the tissue to produce a final stain concentration of 20 μg/ml. The samples were stained for 1 h before microscopy using a Zeiss Axio Imager confocal microscope (Zeiss, Germany) with 488 nm excitation and 510 nm emission wavelength.

**Quantification of fungal biomass in infected tissues.** Quantification of fungal biomass was performed by chitin measurement as described by Ayliffe et al.[95]. Infected leaf tissues from four biological replicates of cv. Bowman and BW746 were harvested at 2, 4, and 8 dpi, weighed, and placed in 15-ml Falcon tubes. One M KOH containing 0.1% Silwet L-77 (Lehle Seeds, U.S.A.) was added to cover the tissue entirely. After autoclaving, the tissues were washed and neutralized as described in the "histological analysis" section. Subsequently, the liquid was poured off and replaced by 1 ml of Tris (pH 7.0) for each 200 mg of plant tissue. The plant tissue was macerated by sonication for 1 min to produce a fine, uniform tissue suspension. Each sample was stained with WGA-FITC (Sigma–Aldrich) dissolved in water by repetitive pipetting before being left to stand for 10 min at room temperature. Samples were then centrifuged at $600 \times g$ for 3 min. The supernatant containing the unbound stain was removed by pipetting, and the pellet was resuspended in 200 μl of 50 mM Tris (pH 7.0). Samples were washed three times in 200 μl of 50 mM Tris (pH 7.0) before resuspension in 100 μl of 50 mM Tris (pH 7.0) and transferred to black, 96-well microtiter trays for fluorometry. Fluorometric measurements were made with a Wallac Victor 1420 multilabel counter (Perkin-Elmer Life Science, U.S.A.) fluorometer with 485 nm adsorption, 535 nm emission wavelength, and 1.0 sec measurement time.

**Pathogen materials.** Four P. hordei pathotypes designated according to the octal notation proposed by Gilmour (1973)[96] (viz. 200 P− [Plant Breeding Institute culture number 518], 5453 P+ [584], 5457 P+ [612], and 5656 P+ [623]) and two P. triticina pathotypes (26-0 [110] and 104-1,2,3,(6),(7),11,13 [547]) were used in this study. Pathotype 5453 P+ was used for phenotyping recombinants and screening mutants, all six pathotypes were used for gene expression, and the first three P. hordei pathotypes were used for multi-pathotype analysis. The suffix P+/P − added to each octal designation indicated virulence/avirulence for resistance gene Rph19[97]. These pathotypes were originally raised from single uredinia on the leaf rust susceptible genotype cv. Gus in the greenhouse and the urediniospores were dried above silica gel for 5–7 days at 12 °C before being stored in liquid nitrogen at the Plant Breeding Institute, the University of Sydney, Australia. Details for each pathotype, including pathogenicity on different resistance genes, are listed in Supplementary Table 9.

**Phenotypic analysis.** At around 8 days after sowing and just prior to second leaf emergence, the seedlings were inoculated with urediniospores suspended in light mineral oil (IsoparL®, Univar, NSW, Australia), at a rate of ~10 mg of spores per 10 ml oil per 200 pots. The suspension was atomized over seedlings in an enclosed chamber using a hydrocarbon propellant at ambient temperatures. The inoculated plants were incubated in a misted dark room (20–22 °C), with mist generated by an ultrasonic humidifier for 18 h, and moved to a temperature-controlled micro-climate room maintained at 23 °C under natural light. The rust responses of at least ten independent seedlings for each line were recorded at 8–10 days post-inoculation using the "0"–"4" infection type (IT) scale[98] with cv. Gus as the susceptible control. IT scores vary from complete immunity "0" without any visible symptoms to full susceptibility "4" with large uredinia without chlorosis. The letters "c", "n" indicated chlorosis or necrosis. The symbols "−" or "+" indicated lower or higher infection types than usual. An IT of 3 or higher was interpreted as susceptible; further details are provided by Park and Karakousis[99].

**Plant materials and growth conditions.** The basic map of the Rph3 locus was generated using 182 recombinant inbred lines (RILs) derived from the cross cvs. Scarlett (Rph3) × Tallon (rph3). Based on the genotypic and phenotypic data (Supplementary Data 1), a subset of 42 lines, one resistant and one susceptible bulk of 10 samples each carrying recombination events adjacent to Rph3, and the two parents were chosen for genotyping using selected tGBS markers. A high-resolution genetic map of the Rph3 locus was constructed based on pooled data for 10,411 $F_2$ plants derived from six segregating populations (Supplementary Table 3). The segregants were genotyped using DNA markers flanking the Rph3 locus (MLOC_005 and MLOC_040). Progeny in which a recombination event had occurred between these markers were further genotyped using internal DNA markers to define the recombination site. All recombinants were self-pollinated to select homozygous recombinants using appropriate DNA markers, and the homozygotes were challenged with Rph3-avirulent P. hordei pathotype 5453 P+ and were scored for rust response based on our phenotyping platform to have unequivocal phenotypic data. Additionally, homozygous recombinants scored for all internal DNA markers.

An international barley collection of 78 accessions representing different sources and alleles of Rph3 based on previous research was subjected to multi-pathotype tests to study the allelic variation (Supplementary Data 6). Each accession has initially been multiplied from a single grain to ensure genetic purity. Genotypic and phenotypic data were collected from each pure line.

**DNA isolation and marker analysis.** $F_2$ grains were sown in 96-punnet (12 × 8) trays filled with potting mix. At the 8-day-old stage after the emergence of the second leaf, about 30 mg of the first leaf of each seedling was sampled into a 96-well collection tube (12 × 8 wells) containing two ball bearings and subjected to DNA extraction using an SDS method. To stabilize the DNA, 450 μl of extraction buffer

including 0.1 M of Tris-HCl buffer (pH 8.0), 0.005 M EDTA buffer (pH 8.0), 0.5 M NaCl, 2-Mercaptoethanol (70 μl/100 ml buffer), and RNAse (100 μg/ml) were added to each sample before crushing. A TissueLyzer II (Qiagen, Germany) at 25 Hz for 2 min was used for crushing the leaf material in the extraction buffer. The final mixture was then added with SDS solution (1.2% final concentration) to solubilize the proteins and lipids at 65 °C for 60 min. The remaining proteins were precipitated by adding ammonium acetate 7.5 M to reach a final concentration of 2 M. The mixture was incubated at 4 °C for 60 min, followed by centrifuging at 4800 rpm (4327 × g) for 10 min to separate debris and the aqueous phase. The upper phase containing genomic DNA was transferred to a new 96-well format plate and pelleted out by adding 100 μl of chilled isopropanol to 100 μl of supernatant. The pellet was twice washed using 100 μl of 70% ethanol before being slowly dissolved in 200 μl TE 0.1× buffer for 6 h for downstream applications.

Primer3Plus software (http://www.bioinformatics.nl/cgi-bin/primer3plus/primer3plus.cgi) was used to design PCR primers that were subsequently synthesized commercially (Sigma–Aldrich, Australia). Each 10 μl PCR contained 0.2 units of high-fidelity DNA polymerase (MyFi™, Bioline, Australia), 0.3 μM of each primer, 1× MyFi reaction buffer (Bioline, Australia), and 20 ng of genomic DNA. Thermocycling conditions consisted of an initial denaturation of 95 °C for 10 min followed by 30 cycles of 94 °C for 30 seconds, 55–60 °C for 30 seconds, 72 °C for 30 seconds, followed by a final extension at 72 °C for 10 min. PCR products were digested (using a suitable endonuclease when required (Supplementary Data 2)) for three hours under the recommended temperature. The digested products were monitored by electrophoresis on an agarose gel and visualized by staining with 6× GelRed® (Biotium, USA) (1.5 μl/100 ml agarose gel).

**Physical mapping**. The sequences of high-confidence genes and non-repetitive sequences on Chr. 7HL extracted from the IPK Barley server (https://webblast.ipk-gatersleben.de/barley_ibsc/viroblast.php) were selected to develop markers to construct a high-resolution map of the Rph3 locus. The sequence of each marker generated for parental lines by Sanger sequencing was used to do homology searches against the Barley Pseudomolecules Masked Apr2016 library to determine its physical position. Parental sequences were aligned with ClustalW within the MEGA-X software[100]. Polymorphic restriction endonuclease sites were identified by the dCAPS tool at http://helix.wustl.edu/dcaps/dcaps.html. The sequences of the closest flanking markers MLOC_190 and MLOC_389 were used to determine the physical window of the Rph3 locus in the genome database of barley cvs. Morex and Barke[41].

**Generation of mutant populations**. Grain of cv. Henley (Rph3) and BW746 (Rph3) were treated with ethyl methane sulphonate (EMS) according to Caldwell et al. (2004)[101] with some modifications. Nine batches of barley grains comprising 1200 and 1500 grains of cv. Henley and BW746 were imbibed in a 2000 ml glass flask filled with one liter of deionized water for four hours at ambient temperature. The water was then replaced by 500 ml of 16 mM EMS (0.2%) solution, and the flask was gently shaken for 20 h at ambient temperatures. After treatment, the grains were extensively washed under running water for two hours. Subsequently, the grains were transferred to trays covered with Whatman paper and placed in a fume hood for slow drying (about 16 h) before sowing. The treated grains were sown directly in the field. After 4 weeks, the seedlings were thinned randomly to about ten plants/meter. Approximately 400 spikes of cv. Henley and 600 spikes of BW746 were harvested from 300 M1 plants each.

**Mutant screening**. In total, 350 and 500 M2 single heads from cv. Henley and BW746, respectively, were used for gene validation. The M2 spikes and selected M3 families were screened for knockout mutants using the Rph3-avirulent pathotype 5453 P+. Each M3 line was sown in an independent pot and tested for rust response. All the susceptible and three resistant plants were transplanted for each family showing a segregating reaction. The Rph3 locus (8519 bp in length) in M3 susceptible plants was resequenced using the Sanger method. The M3-derived M4 families were progeny-tested to confirm the phenotype of M3 plants.

**Allelism test**. The EMS-induced mutants were divided into two groups. Group I included mutants with nucleotide changes within the Rph3 locus, and Group II consisted of mutants with no nucleotide change within the locus (8.5 kb). Three types of crosses were made to test the allelism of the EMS-induced mutations: Group I × Group I; Group I × Group II; Group II × Group II. The F1 grains and their parents were inoculated with P. hordei pathotype 5453 P+ to test the allelic status of these mutants (Supplementary Table 5).

**Rph3 construct**. A genomic DNA segment of 7096 bp sequence including 2196 bp sequence of the gene Rph3, 3400 bp of upstream sequence including the 5′-UTR region, and 1500 bp downstream sequence following the stop codon including the 3′-UTR region (Supplementary Fig. 5) was synthesized in three pieces that were then assembled in a pUC vector. The complete sequence was then cloned along with the nos terminator into binary vector P6oUZm via the SfiI cloning site to form the pRph3::Rph3 construct named D657-Rph3-oex that was eventually used to transform barley.

**Barley transformation**. The plasmid D657-Rph3-oex was introduced into Agrobacterium tumefaciens strain AGL-1[102] using a Gene Pulser II according to the manufacturer's instructions (BioRad, Hercules, CA). Then, agro-inoculation of immature embryos of barley cv. Golden Promise and co-culture of the explants with Agrobacterium was conducted, followed by callus induction and shoot regeneration on hygromycin-containing media as previously described by Hensel et al.[103]. After roots had emerged, planlets were transferred to the soil and grown to maturity. To confirm the presence of T-DNA in candidate transgenic plantlets, genomic DNA was extracted according to Palotta et al.[104], and PCR was performed using primers specific for the junction between CaMV 35S promoter (5′-CATTG GTGGAGCAGCACACTCTC-3′) and hpt gene (5′-GATTCCTTGCGGTCCGAA TG-3′). In addition, the presence of the dominant Rph3 allele was confirmed in each plant by PCR using the specific marker MLOC_400 (forward: 5′-ACGTGA ATGAAATCCGGTTC-3′ and reverse: 5′-GTGCTGCTCTCCGTTGTGT-3′) (Supplementary Fig. 5, Supplementary Data 2).

**Haplotype analysis**. The genomic region covering 8.5 kb of the Rph3 locus was divided into fragments of 5465 and 4882 bp with 1428 bp overlapped for amplification. These fragments were amplified using primer pairs (5 kb_C2 and 5 kb_C5) (Supplementary Data 5), employing LongAmp® Taq DNA polymerase (New England BioLabs, USA) and MyFi™ DNA polymerase (Bioline, Australia) respectively, in a T100 Thermal Cycler (Bio-Rad). To amplify the 5465 bp DNA fragment, each 10 μl PCR contained 0.1 units of LongAmp Taq DNA polymerase, 0.4 μM of each primer, 1× LongAmp Taq reaction buffer, 10 μM of each dNTP, and 50 ng of genomic DNA. The PCR was run with the block preheated to 94 °C before thermocycling. The thermocycling conditions were an initial denaturation of 95 °C for 5 min followed by 30 cycles of 94 °C for 30 seconds, 60 °C for 30 seconds, 65 °C for 6 min, and a final extension at 65 °C for 10 min. The components and thermocycling conditions to amplify 4882 bp were the same as described in the "DNA marker analysis" section but with the elongation step lasting for 5 min instead of 30 seconds. The amplicons were purified using AMPure XP magnetic beads (Beckman Coulter Life Sciences, USA). The sequencing template was subjected to Sanger sequencing using 28 internal primers (14 forward and 14 reverse primers) (Supplementary Data 5) that were designed from the reference DNA sequence of cv. Barke.

**Allelic variation**. The 78 barley accession core collection was challenged with P. hordei pathotypes, 200 P−, 5453 P+, and 5457 P+. Pathotype 5457 P+ differs from pathotype 5453 P+ only in being virulent on Rph3 and is considered a single-step mutational derivative of the latter[105,106]. Alleles of Rph3 conferring resistance and susceptibility were differentiated using the co-dominant cleaved amplified polymorphic sequence (CAPS) marker MLOC_198, which was completely linked to Rph3 in the high-resolution map (Supplementary Data 6). The dominant marker Rph3_full covering the full-length sequence of Rph3 (all exons and introns) was used to detect the presence/absence of the Rph3 segment in these accessions. A total of 41 barley accessions postulated to carry different alleles of Rph3 based on infection type array were subjected to Sanger sequencing of the 8519 bp interval as described in the previous section. The sequences were aligned using the MUSCLE function at (https://www.ebi.ac.uk/Tools/msa/muscle/) to find any variation.

**The frequency of Rph3 in a diverse barley collection**. Genotype-by-sequencing (GBS) was previously applied to a diverse collection of elite, landrace, and wild barley accessions (n = 22,942) by digesting genomic DNA using PstI and MspI endonucleases and sequenced using an Illumina HiSeq2500[61,62]. GBS sequencing data were downloaded from NCBI for 22,628 barley accessions (PRJEB8290, PRJEB23967, PRJEB24563, PRJEB24627, and PRJEB26634). Raw GBS sequencing data for the Wild Barley Diversity Collection (n = 314) from Sallam et al. 2017[62] was provided by Prof. Brian Steffenson (University of Minnesota). Sequencing data from cvs. Morex and Barke were initially mapped using BBmap (v38.86) to identify regions encompassing GBS markers with parameters of a minimum identity of 95% and maximum InDel of 5 bp. Two adjacent GBS markers (gRph3_I1E2 and gRph3_E2I2) mapped to the region encompassing intron 1, exon 2, and intron 2 of Rph3. Genomic regions with GBS markers were used as a template for k-mer analysis using sect in the k-mer analysis toolkit (KAT; https://github.com/TGAC/KAT) with k = 27[107]. For every accession, the number of non-zero k-mers was used as a metric for the presence or absence of the Rph3 haplotype based on the cv. Barke genomic sequence. A bimodal distribution was identified among sequenced accessions, and a threshold of 158 k-mers was used to classify for the presence or absence of the Rph3 allele. A dominant PCR marker for Rph3 (forward: ACGT-GAATGAAATCCGGTTC; reverse: GTGCTGCTCTCCGTTGTGT) was used in multiplex with primers on a BAC end sequence (0206D11_T7) from the Mla locus that amplified universally (forward: CTGGTTTGTTGTTGCTATGCGTTG; reverse: TCATTTGGTGTGTGGGGCAAAG)[108]. PCR was performed using GoTaq DNA Polymerase (Promega) in 25 μl reactions following the manufacturers' protocol. The thermocycling conditions consisted of initial denaturation of 95 °C for 2 minutes followed by 35 cycles of 94 °C for 30 seconds, 60 °C for 30 seconds, 72 °C for 35 seconds, and a final extension at 72 °C for 5 minutes.

**Sequence annotation**. The *Rph3* allele was extracted from the 8.5 kb DNA sequence delimited by markers MLOC_190 and MLOC_389 in cv. Barke genome. The 8.5 kb region without any annotated genes or repetitive elements was processed by the gene structure prediction program FGENESH (http://www.softberry.com/berry.phtml) using the monocotyledonous plant codon usage matrix[109] as a reference.

**Promoter analysis**. The promoter of *Rph3* was defined as the sequence used for transformation prior to RNA-seq evidence of transcription. Pairwise alignments of *Rph3* promoters in Barke (*Rph3*) and Morex (*rph3*; HORVU.MOREX.r3.7HG0741050) were performed using Geneious alignment (gap open penalty of 12; gap extension penalty of 3). Putative transcription factor binding sites were obtained from PLACE (version 30.0)[110] and identified in the promoter sequences based on regular expression analysis.

**Prediction of secondary protein structure**. After confirming the full-length cDNA using RNA-Seq data, the putative amino acid sequence of RPH3 protein was used in a homology search against the Conserved Domain Database (CDD v3.18 - 55570 PSSMs) on the NCBI website (https://www.ncbi.nlm.nih.gov/Structure/cdd/wrpsb.cgi) with a default expected value threshold of 0.01. Secondary structure prediction was performed using three independent online tools, including Protter (http://wlab.ethz.ch/protter/start), TMPRED (https://embnet.vital-it.ch/software/TMPRED_form.html), and TMHMM (http://www.cbs.dtu.dk/services/TMHMM/). Simultaneously, an HHPred search was performed online (https://toolkit.tuebingen.mpg.de/tools/hhpred) using the MPI Bioinformatics Toolkit[111] with the amino acid sequence of the RPH3 protein as query to search for structurally similar proteins. The search used all default parameters. The hit with the highest secondary structure score with RPH3 was used as a template to generate alignment in the PIR format that then was fed to MODELLER to visualize the secondary structure of the RPH3 protein.

**Evolution history of RPH3 in cereals**. The cDNA sequence of the *Rph3* allele was used as a query in the BLASTX function on NCBI to find similar proteins. The *Rph3* coding sequence was used as a query for the BLASTN function to search for homologs and orthologs in barley (https://webblast.ipk-gatersleben.de/barley_ibsc/), oat (https://avenagenome.org/), wheat (https://urgi.versailles.inra.fr/blast/?dbgroup=wheat_iwgsc_refseq_v1_chromosomes&program=blastn), and in 24 other monocot species available in ensemble plants (https://plants.ensembl.org/Multi/Tools/Blast). In parallel, the search for sequence homologs was performed using profile Hidden Markov Models (https://toolkit.tuebingen.mpg.de/tools/hmmer)[111] using all default parameters with the amino acid sequence of the Rph3 protein as a query. The duplicated subjects were removed before phylogenetic analysis. Amino acid sequences of RPH3 and its homologs/paralogs were used as an entry in SALAD (https://salad.dna.affrc.go.jp/salad/en/) to analyze their motif composition.

**Phylogenetic analysis**. The phylogenetic tree was constructed by the maximum likelihood using BEAST v1.10.4. Sequences were aligned using Clustal omega in EMBL-EBI (https://www.ebi.ac.uk/Tools/msa/clustalo/) before a setting step was conducted using the software BEAUti v1.10.4. Substitution model "Blosum62"[112] was chosen by the software based on the imported sequences. Other options were selected including the "Speciation: Birth-death Process" model for Tree prior[113] as the dataset contained a mixture of within- and between-species sequences, "Uncorrelated relaxed clock" for clock type with lognormal related distribution[114], and "10,000,000" for the MCMC value. The sampling frequency was set to 1,000 to have 10,000 samples recorded. The software TreeAnnotator v1.10.4 was used to create the consensus tree that was visualized by FigTree v1.4.4. The posterior value demonstrated the likelihood of each branch.

**Gene expression analysis by RT-qPCR**. The first leaves of the inoculated plants from the lines used in this study were harvested at different timepoints with three biological replicates, flash-frozen in liquid nitrogen, and then stored at −80 °C until RNA extraction. Total RNA was extracted from the samples using TRIzol$^{TM}$ Reagent (Thermo Fisher Scientific Ltd) following the manufacturer's instructions. The genomic DNA was digested using DNase I (Sigma–Aldrich). RNA quality was checked on agarose 1.5% gels, and quantity was reviewed on a NanoDrop$^{TM}$ 1000 Spectrophotometer (Thermo Fisher Scientific Ltd). RT-qPCR was performed using Luna® Universal One-Step RT-qPCR Kit (New England Biolabs®) following instructions from the manufacturer and the CFX$^{TM}$ Real-Time PCR Detection System (Bio-RAD). ADP-Ribosylation Factor (*ADPRF*) was used as a reference gene, and RT-qPCR data were analyzed using the ΔΔCq method. Primer sequences for RT-qPCR are listed in Supplementary Table 10.

**RNA sequencing and data analysis**. One set of first leaf seedlings of cv. Bowman and BW746 was inoculated with *P. hordei* pathotype 5453 P+, and a second set was used for mock inoculation. The first leaves from both treatments were harvested at two dpi and flash-frozen in liquid nitrogen until RNA extraction. Twelve samples (two genotypes, two treatments/genotype, and three biological

replications/treatment) were subjected to total RNA extraction using a Spectrum$^{TM}$ Plant Total RNA kit (Sigma–Aldrich) following the manufacturer's instructions. The RNA samples were initially quantified using a NanoDrop$^{TM}$ 1000 spectrophotometer (Thermo Fisher Scientific Ltd), degradation and potential contamination were checked on 1.5% agarose gel, and RNA integrity and quantitation were measured using an Agilent 2100 analyzer. A library was prepared using the NEBNext® Ultra$^{TM}$ RNA Library Prep Kit. RNA sequencing was conducted using an Illumina PE150 that generated 40 million paired-end reads for each sample. Individual RNA-Seq datasets were assessed for quality using FastQC (0.11.9)[115]. Trimmomatic (v0.39) was used for trimming reads using parameters 'ILLUMINACLIP:TruSeq3-PE.fa:2:30:10 LEADING:5 TRAILING:5 SLIDINGWINDOW:4:15 MINLEN:36'[116]. Pseudoalignments using Kallisto (v0.46.0)[117] were made using the barley transcriptome (high and low confidence gene models) based on the 2017 genome annotation[118] and the transcript sequence of *Rph3*. Differential gene expression analysis was carried out using DESeq2 (1.20.0) with default parameters[119]. The false discovery rate was controlled at 5% (*q*-value of 0.05). Gene ontology analysis was performed using g:Profiler. RNA-seq data have been deposited in NCBI SRA in BioProject PRJNA731362.

**Gene cloning**. To generate *Rph3*$^{WT}$, *Xa10* (AGE45112), *Xa23* (AIX09985), *Xa27* (AAY54165), and *Bs4C* (AFW98885) expression constructs, the corresponding coding sequences were synthesized and cloned with the Twist Bioscience's clonal gene synthesis service, using codon optimization for expression in *N. benthamiana*, and removal of the *Bsa*I and *Bpi*I internal restriction sites. The coding sequences were cloned into the pTwist-Kan-High-copy vector, including two flanking *Bsa*I restriction sites for subsequent Golden Gate cloning. The resulting plasmids were used in the Golden Gate assembly with pICH85281 (*mannopine synthase* + Ω promoter (*Mas Ω*), Addgene no. 50272), pICSL50009 (6xHA, TSL Synbio), pICSL60008 (Arabidopsis heat shock protein terminator, HSPter, TSL Synbio), and the binary vector pICH47732 (Addgene no. 48000). The Rph3$^{L93F}$ and Rph3$^{P126L}$ mutants were generated by PCR site-directed mutagenesis using Phusion High-Fidelity DNA Polymerase (Thermo Fisher), with pTwist-Kan-High-copy::Rph3$^{WT}$ as a template. The internal primers flanking the mutation sites Rph3_L93F_fw (5′-CACAACGCATtTTAACATGAATAG), Rph3_L93F_rv (5′-CTATTCATGTTAAaATGCGTTGTG), Rph3_P126L_fw (5′-GAATGGTGATCCtTAAGGATCATTC), and Rph3_P126L_rv (5′-GAATGATCCTTAaGGATCACCATTC), along with the outermost flanking primers Rph3_fw (5′-aaGAAGACaaAATGGATGCCGGAGCTTTTG) and Rph3_rv (5′-aaGAAGACaaCGAAccTGCCAGCACTACAAC), were used to generate single PCR fragments upstream and downstream of each mutation site. The purified fragments were fused by PCR using primers Rph3_fw and Rph3_rv. The resulting full-length fragments were cloned into the pICSL01005 vector (TSL Synbio) using Golden Gate assembly. PCR amplification of the Rph3$^{E72*}$ truncated mutant was done using primers Rph3_fw and Rph3_E72*_rv (5′-aaGAAGACaaCGAAccGGAGCCCTTTGTCTGAACGG). The resulting fragment was purified and used in a Golden Gate assembly with the pICSL01005 vector (TSL synbio). These assemblies were used for subsequent Golden Gate cloning into binary vectors for transient expression in a similar assembly reaction as described for Rph3$^{WT}$. In all cases, the mutants were verified by DNA sequencing. *Escherichia coli* DH5α was used for molecular cloning experiments.

**Transient gene expression and cell death assays**. *N. benthamiana* plants for transient gene expression assays were grown in a growth chamber held at 22–25 °C with 45–65% humidity and 16/-8 h light-dark cycle. Transient expression in *N. benthamiana* was performed by infiltrating leaves of four-week-old plants with *A. tumefaciens* GV3101 pMP90 carrying a binary expression plasmid containing the coding sequence of the protein of interest. Bacterial suspensions were prepared in infiltration buffer (10 mM MES, 10 mM MgCl$_2$, and 150 mM acetosyringone) and adjusted to an OD$_{600}$ = 0.4. Leaves were harvested and imaged three days post infiltration. Each experiment was performed three times, infiltrating two leaves of 3–4 plants each time.

**Protein immunoblotting**. To assess protein accumulation, *A. tumefaciens* suspensions carrying the constructs of interest were adjusted to an OD$_{600}$ = 0.2 in infiltration buffer (10 mM MES, 10 mM MgCl$_2$, and 150 mM acetosyringone) and infiltrated 4-week-old *N. benthamiana* plants. Six 8 mm leaf disks were harvested two days post-agroinfiltration, and snap-frozen in liquid nitrogen. Leaf tissue was ground and homogenized in extraction buffer (10% glycerol, 50 mM Tris-HCl (pH 7.5), 1 mM EDTA, 150 mM NaCl, 2% IGEPAL (SIGMA), protease inhibitor (complete, mini, EDTA-free, Sigma # 11836170001, 1 tablet/50 mL), 1 mM Na$_2$MoO$_4$*2H$_2$O, 1 mM NaF, 1.5 mM Na$_3$VO$_4$, 5 mM dithiothreitol (DTT), 1 mM phenylmethylsulfonyl fluoride (PMSF)). Samples were incubated at 4 °C for 30 min, and centrifuged at 13,000 × *g* for 10 min at 4 °C. The supernatant was collected and centrifuged again to remove plant debris. The plant extract was mixed with Laemmli sample buffer (Bio-Rad) and boiled at 95 °C for 10 min. Samples were separated by SDS-PAGE and transferred onto a PVDF membrane using the Trans-Blot turbo system (Bio-Rad). The membrane was blocked with 5% skimmed milk powder in TBST for a minimum of 1 hour at room temperature. HA epitope-tagged proteins were detected with Anti-HA, peroxidase, and high-affinity antibody (Roche, 12013819001) in a 1:5000 dilution in 5% skimmed milk powder

in TBST. The signal was visualized using Pierce™ ECL Western Blotting Substrate (Thermo Fisher Scientific) with 50% SuperSignal™ West Femto Maximum Sensitivity Substrate (Thermo Fisher Scientific). Membrane imaging was performed with an Amersham ImageQuant 800 western blot imager system (Cytiva). Equal loading was checked by staining the PVDF membrane with Ponceau S solution (Sigma, #6226-79-5).

**Reporting summary**. Further information on research design is available in the Nature Research Reporting Summary linked to this article.

## Data availability

RNA-seq data have been deposited in Sequence Read Archives at National Center for Biotechnology Information (NCBI) under BioProject accession number PRJNA731362. The full-length cDNA and genomic sequence of the Rph3 gene have been deposited in NCBI with the accession number MZ561688 and MZ561689. Source data are provided with this paper.

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

## Acknowledgements

We thank Prof. R. A. McIntosh for suggesting the experiment that assessed the tissue specificity of *Rph3* expression; Dr. P. Zhang for help in mutagenesis using grains; Dr. C. Dong and Ms. M. Demers for assistance with RNA extraction; Dr. E. Wang for help with RT-qPCR; Dr. P. Dracatos for discussion related to mutant analysis; Dr. E. Lagudah for discussion on protein function; Prof. S. Ho for suggestions related to phylogenetic analysis; Mr. M. Williams for technical support with plant growth facilities; Mrs. S. Sommerfeld and Mr. R. Hoffie for assistance in generating and allocating transgenic barley; Mrs. K. Niedung for technical support with artificial inoculation, DNA extraction and PCR on transgenic materials; Dr. I Hernández-Pinzón for technical support with PCR on wild barley germplasm; GRDC, Gatsby Foundation, UKRI-BBSRC (BBS/E/J/ 000PR9795), and BMBF project (031B0199B) for financial support; and the Australian Awards Scholarship for providing financial support to HD.

## Author contributions

R.F.P. and M.P. conceived the project. R.F.P. provided all rust isolates and information on pathogenicities and oversaw all rust phenotyping. M.A. and H.X.D. conducted histology experiments and chitin assays by confocal microscopy. H.X.D. and M.P. constructed the genetic and high-resolution maps. N.S. and M. Mascher provided the reference sequence of the *Rph3* locus, M.P. and M. Mascher annotated the final locus of *Rph3*. H.X.D. and M.P. created mutant materials, H.X.D. and D.S. screened the knockout mutants. H.X.D., D.S., R.F.P., and M.P. designed and performed the haplotype analysis. H.X.D., R.F.P., and D.S. performed a multi-pathotype test and gene postulation. H.X.D. and M.P. performed the expression analysis using RT-qPCR. M.P. and M. Moscou designed the RNA-Seq experiment, H.X.D. experimented, M. Moscou analyzed data. H.X.D. and M.P. performed phylogenetic analysis. M.P. designed the transgenic construct, G.H. and J.K. created the transgenic material, and D.P. tested transgenic progeny using I-16 isolate. M. Moscou and D.G. designed and performed transient expression analysis. M. Moscou analyzed the origin and frequency of the *Rph3* allele in the barley gene bank. M.P. and R.F.P. supervised the project. H.X.D. and M.P. wrote the manuscript. All authors reviewed and commented on the manuscript.

## Competing interests

The authors declare no competing interests
