## [Peer Review File · Nature Communications]

The barley leaf rust resistance gene Rph3 encodes a unique membrane-binding protein and is induced upon infection by avirulent pathotypes of *Puccinia hordei*REVIEWER COMMENTS

Reviewer #1 (Remarks to the Author):

This paper describes the molecular isolation of the barley Rph3 leaf rust resistance gene and its functional and phylogenetic analysis. Rph3 was isolated by map-based cloning and validated by mutant analysis and transformation. The identified gene sequence shows little or no similarity to any known protein but is predicted to encode a transmembrane protein which is partially conserved within other Triticeae species. Within the tested barley gene pool there was no diversity. Importantly, the gene is only and specifically induced and expressed after infection with avirulent barley leaf rust isolates. The authors conclude that Rph3 encodes an “executor” protein which is specifically induced by AvrRph3, either directly or indirectly. Overexpression of Rph3 in *Nicotiana* resulted in cell death, very similar to overexpression of known executor proteins encoded by resistance genes in rice and pepper against *Xanthomonas* bacterial pathogens.

The identification and characterization of Rph3 reveals a type of plant protein not known to be involved in disease resistance, revealing a potentially novel molecular mechanism. The well studied Xa or Bs resistance /executor proteins are not sequence-related to Rph3 and confer resistance against bacterial diseases, whereas the work here refers to resistance to a fungal pathogen. The work presents convincing data on the identity of the gene and reveals some functional and phylogenetic aspects that are of great interest. I have some comments and suggestions:

Major points:

L64 – 66: “Ten genes have been cloned, six encode NLRs. The three remaining genes...” This does not add up to 10. Furthermore, the authors cite a recent paper in *Nature Communication* (ref 15) on the cloning of Lr14a, but they do not mention that it encodes an ankyrin-repeat transmembrane protein with similarities to Ca²⁺ permeable non selective cation channels. In fact, Lr14a is also a gene that is specifically induced by avirulent leaf rust isolates only.

Similarly, the discussion (L383) fails to address Lr14a.

L107: It is not clear if the 99% refer to actually determined content based on SNP analysis, or if this a theoretical value.

It needs to be shown that the putative splice site mutant M466 indeed results in an aberrant splicing of the gene.

L202: The authors used blast searches to identify homologous genes. This is not very sensitive: It was recently found for the wheat LR14a protein that similarities with potential cation channels were only detected using a Hidden Markov model such as HHPred. The LR14a homolog ACD6 in Arabidopsis was also specifically identified as a Ca²⁺ channel only by Hidden Markov analysis (<https://www.biorxiv.org/content/10.1101/2021.01.25.428077v1.abstract>). Therefore, the authors should use more sophisticated approaches than blast to identify putative proteins with structural similarities to RPH3.

It must be specified what the MasDelta promoter refers to. The reference in Materials and Methods to Addgene results in a plasmid with normal mas promoter. What is the delta version used here? This is particularly important as the use of this promoter for overexpression of Xa27 in *N. benthamiana* resulted in data which are in conflict with the literature. Legend of Figure 4 then mentions only the mas promoter. Is this the delta version?

The *N. benthamiana* experiments used HA tagged protein variants. Protein expression levels should be shown in a Western blot.

The sub-chapter starting L262 needs clarification of description and data analysis. It is not stated if an avirulent isolate was used, which seems to be the most important information for such an experiment. Given the fact that Rph3 is only induced by an avirulent isolate, I would have expected to see a transcriptomic analysis of an Rph3 containing cultivar with a virulent and avirulent isolate, respectively. The analysis and focus in the current version of the manuscript fails to address this most important aspect.

There is no sequence variability of the Rph3 haplotype in cultivated barley (L294). However, some novel haplotypes were identified in wild barley and additional allelic variants of Rph3 are suggested (L314). The manuscript should include the Rph3 sequence of at least two genotypes with suspected Rph3 variants to clarify and complete the diversity analysis.

Figure S1A: the quality of the figure is insufficient to review it. There is more growth in genotype BW746?

Minor points, including typos

L98: "amino acid" not "mino acid"

L117: Should be Figure 1c, not 2c

L358: rephrase. It is a bit awkward to state that promoter sequences target effector proteins.

L396: replace "to triggers" by "to trigger"

L453 This paragraph is mostly a repetition and can be deleted or shortened.

Reviewer #2 (Remarks to the Author):

The MS presents support for the identification of a new class of R gene in barley to rust fungi. The R gene (Rph3) is race-specific, meaning some isolates of the pathogen are virulent in the presence of the gene. The hypothesized effector of the pathogen has not been isolated at this point. The novelty of the gene is the protein product and the apparent requirement for inducible expression upon challenge by the avirulent pathogen and not the virulent pathogen. An exhaustive set of approaches to demonstrate the validity that so-called ORF1 is the R gene was performed: Fine scale mapping, EMS mutagenesis, comparisons of R-specific germplasm, and gene transfer to a susceptible line of barley. In addition, the product elicits a hypersensitive reaction with transiently expressed in *Nicotiana benthamiana*. The authors also present an extensive analysis of related coding sequences in related species. Comparisons are made to so-called executor R genes of rice and pepper, which are targeted and induced by effectors that act as transcription factors. Speculation is provided that the cognate effector may function in a similar fashion.

The new class is an exciting find and presents an occurrence of a

R gene directed to a fungus that requires induction of expression. The EMS mutagenesis and gene transfer provide strong evidence that the ORF2 is the gene. The results are of considerable significance.

The main criticism/suggestions of the work is the length and lack of basic promoter analysis given the proposed requirement for pathogen-specific gene induction. More to the point, three items would enhance the manuscript.

1. The manuscript could be shortened considerably. The discussion is very long given little analysis of the mode of gene induction. The discussion of TALE-mediated expression presents too much detail, for example, given the speculative nature of the hypothesis.

2. The authors should measure the expression of Rph3 in the transgenic progeny.
3. An analysis of the promoter region and comparisons of the promoter region in the various germplasm and lines should have been presented with the protein analysis since much discussion in the MS was devoted to the requirement for gene induction. Many aspects for promoter function can be obtained by comparative and consensus analyses.

Minor comments:

4. line 98 typo mino to amino
5. In discussion, ABA has documented effects both for enhanced susceptibility upon expression or resistance upon suppression. (As noted above, this aspect of discussion can be removed as very speculative. Homology to portions to NCED, which is related to larger set of proteins does not readily translate to increased ABA.)
6. The reference list is very large, and, again, could be reduced by reduction of the discussion.
7. It is somewhat unfortunate the name "executor" is used for two reasons. One, the "executor" comparison is still speculative at this point in regards to details, and, two, the term probably should be "executioner". But be that as it may, the term has been adopted. Cannot the title just claim a novel R gene.

Reviewer #3 (Remarks to the Author):

The manuscript entitled "The barley leaf rust resistance gene Rph3 encodes a putative executor protein" by Dinh et al describes molecular cloning and characterization of a putative new executor R gene Rph3 from barley. This is a very interesting work because the R gene Rph3 confers resistance to barley leaf rust caused by fungal pathogen *Puccinia hordei*, not the bacterial pathogen *Xanthomonas* spp. that are specifically associated with the so-called executor R genes, previously.

The Rph3 (Reaction to *Puccinia hordei* 3) locus was first discovered in barley landrace 'Estate' using classical genetics in 1967. Although pathotypes with virulence for Rph3 have been detected in all barley growing areas, Rph3 remains a valuable source of resistance for barley breeding. The authors of this manuscript isolated the Rph3 gene by map-based cloning, confirmed by mutational analysis and transgenic complementation; the conclusion on Rph3 cloning was strongly supported by abundant data.

What's interesting is that the Rph3 gene encodes a unique transmembrane resistance protein differing from all known plant disease resistance proteins at the amino acid sequence level. The authors clearly demonstrated that the Rph3 gene was expressed only in interactions with Rph3-avirulent isolates of *Puccinia hordei*, a phenomenon observed for the transcription activator-like effector-dependent executor R genes conferring resistance to *Xanthomonas*. The Rph3 gene also possesses some other features of the known executor R genes isolated from pepper and rice, such as encoding a protein with transmembrane-helix domains, expression induces cell death response in *Nicotiana benthamiana* as well as in host plants. Based on these observations, the authors concluded that 'executor' genes exist in the Triticeae tribe, and executor genes (e.g., Rph3) could confer resistance against fungal pathogen.

This work is important because it put new insights to better understanding the scope and resistance mechanism of executor R genes, as well as facilitating the application of the valuable R gene Rph3 in barley breeding for leaf rust resistance.

Comments and suggestions:

1. The major limitation of this work is the lack of information about the cognate effector (AvrRph3) in *P. hordei*, which should specifically activate expression of Rph3. This effector is a critical factor in determining Rph3 as an executor gene.
2. If possible, provide the evidence of mis-spliced RNAs in the M466 mutant.
3. The result section "Transcription dynamics of Rph3-mediated resistance at two days post-inoculation" does not provide explicit genes associated with pathways or mechanism of Rph3-mediated resistance. It could be moved into discussion.
4. Line 117: "Figure 2c)" should be Figure 1c.
5. Line 122: Better to change the "higher response" as "higher susceptibility".
6. Line 621-623: The 7,096 bp genomic DNA segment for Rph3 construct was synthesized? Please provide more information about the synthesis.
7. Line 1116/Fig.SI1: It seems Bowman and BW746 were wrongly positioned?
8. Line 1216-1218/Fig.SI14: The mentioned black/green/black arrows can't be found in the figure?

1. Reviewer #1:

Comment: L64 – 66: “Ten genes have been cloned, six encode NLRs. The three remaining genes...” This does not add up to 10. Furthermore, the authors cite a recent paper in Nature Communication (ref 15) on the cloning of *Lr14a*, but they do not mention that it encodes an ankyrin-repeat transmembrane protein with similarities to Ca²⁺ permeable nonselective cation channels. In fact, *Lr14a* is also a gene that is specifically induced by avirulent leaf rust isolates only.

Response: We thank Reviewer 1 for the constructive comments. The word “three” has been changed to “four” that was a typing error. We have added “and a membrane-bound ankyrin repeat protein” in page 3, line 69 of the revised version to specifically address this comment. The expression profile of *Lr14a* has been added into “The inducible expression of *Rph3*” section in page 12, lines 381 of the revised version.

Comment: Similarly, the discussion (L383) fails to address *Lr14a*.

Response: The following phrase has been added to the “Molecular function of the RPH3 protein” section in page 13, lines 423 - 424 of the revised version: “, and leaf rust resistance gene *Lr14a* from hexaploid wheat encodes protein with twelve ankyrin repeats”.

Comment: L107: It is not clear if the 99% refer to actually determined content based on SNP analysis, or if this a theoretical value.

Response: The word “theoretically” added in page 8, line 109 of the revised version to address this comment.

Comment: It needs to be shown that the putative splice site mutant M466 indeed results in an aberrant splicing of the gene.

Response: A sentence has been added in page 6, lines 170 - 171 of the revised version. “The mutant line M466 did not survive, and the changes in the protein structure of the RPH3 protein could not be examined.”

Comment: L202: The authors used blast searches to identify homologous genes. This is not very sensitive: It was recently found for the wheat *LR14a* protein that similarities with potential cation channels were only detected using a Hidden Markov model such as HHPred. The *LR14a* homolog *ACD6* in *Arabidopsis* was also specifically identified as a Ca²⁺ channel only by Hidden Markov analysis (<https://www.biorxiv.org/content/10.1101/2021.01.25.428077v1.abstract>). Therefore, the authors should use more sophisticated approaches than blast to identify putative proteins with structural similarities to RPH3.

Response: This comment is highly appreciated. We have performed the Hidden Markov analysis and the text of the analysis using HHpred model was added in page 8, lines 225 - 233 and one sentence “The analysis using a Hidden Markov Model HHMER with the RPH3 amino acid sequence as query showed seven hits that are consistent with the results given by BLASTX search.” was added in page 8, lines 238 - 240 of the revised version.

Comment: It must be specified what the MasDelta promoter refers to. The reference in Materials and Methods to Addgene results in a plasmid with normal mas promoter. What is the delta version used here? This is particularly important as the use of this promoter for overexpression of Xa27 in *N. benthamiana* resulted in data which are in conflict with the literature. Legend of Figure 4 then mentions only the mas promoter. Is this the delta version?

Response: We apologise for the confusion on this, the delta was meant to be omega, but was changed in the transfer of text. This promoter is listed as a module that is publicly available as described in the Materials and Methods: “The resulting plasmids were used in the Golden Gate assembly with pICH85281 (mannopine synthase + Ω promoter (Mas Ω), Addgene no. 50272), pICSL50009 (6xHA, TSL Synbio), pICSL60008 (Arabidopsis heat shock protein terminator, HSPter, TSL Synbio), and the binary vector pICH47732 (Addgene no. 48000).” We will supply sequences for plasmids as part of the figures Source Data.

Comment: The *N. benthamiana* experiments used HA tagged protein variants. Protein expression levels should be shown in a Western blot.

Response: Protein expression levels were shown in Figure SI15 and cited in page 9, line 288 of the revised version. The detailed procedure of the Western blot experiment was added to the SI Materials and Methods section, page 25, lines 831 - 847 of the revised version.

Comment: The sub-chapter starting L262 needs clarification of description and data analysis. It is not stated if an avirulent isolate was used, which seems to be the most important information for such an experiment. Given the fact that Rph3 is only induced by an avirulent isolate, I would have expected to see a transcriptomic analysis of an Rph3 containing cultivar with a virulent and avirulent isolate, respectively. The analysis and focus in the current version of the manuscript fails to address this most important aspect.

Response: We have added “Rph3-avirulent *P. hordei* pathotype 5453 P+ ...” in Page 9, line 294 of the revised version for clarification. We designed the current experiment with the use of a single avirulent isolate. We focused this experiment on identifying genes that are specifically induced or suppressed in steady-state levels of expression due to having *Rph3* or not (Bowman). We consider this experiment exploratory, as we agree with the reviewer that a full understanding of the down-stream regulatory processes mediating *Rph3*-mediated resistance would need the use of virulent isolates, as well as multiple time points. This becomes a large experiment, which we believe is beyond the scope of the current manuscript.

Comment: There is no sequence variability of the Rph3 haplotype in cultivated barley (L294). However, some novel haplotypes were identified in wild barley and additional allelic variants of Rph3 are suggested (L314). The manuscript should include the Rph3 sequence of at least two genotypes with suspected Rph3 variants to clarify and complete the diversity analysis.

Response: PCR amplification failed for these variants and data is added in page 11, lines 347 - 349 and the Figure S19 to the supplementary section of the revised version.

Comment: Figure S1A: the quality of the figure is insufficient to review it. There is more growth in genotype BW746?

Response: To address this comment, we have replaced the figure with a better quality, changed the figure legend, and changed the description in the result section, page 5, lines 116 - 131 of the revised version.

Comment: L98: “amino acid” not “mino acid”

Response: “mino acid” has been replaced with “amino acid” in page 4, line 100 of the revised version.

Comment: L117: Should be Figure 1c, not 2c

Response: “2c” has been changed into “1c” in page 5, line 116 of the revised version.

Comment: L358: rephrase. It is a bit awkward to state that promoter sequences target effector proteins.

Response: The sentence has been rephrased according to this comment in page 13, line 397 of the revised version. “... plants evolved resistance genes with promoter sequences that are targeted by effector proteins ... ”

Comment: L396: replace “to triggers” by “to trigger”

Response: This has been changed according to reviewer’s comment in page 14, line 436 of the revised version.

Comment: L453 This paragraph is mostly a repetition and can be deleted or shortened.

Response: The repetitions have been removed in page 15, lines 497 - 507 in the revised version.

Reviewer #2 (Remarks to the Author):

Comment: The MS presents support for the identification of a new class of R gene in barley to rust fungi. The R gene (*Rph3*) is race-specific, meaning some isolates of the pathogen are virulent in the presence of the gene. The hypothesized effector of the pathogen has not been isolated at this point. The novelty of the gene is the protein product and the apparent requirement for inducible expression upon challenge by the avirulent pathogen and not the virulent pathogen. An exhaustive set of approaches to demonstrate the validity that so-called ORF1 is the R gene was performed: Fine scale mapping, EMS mutagenesis, comparisons of R-specific germplasm, and gene transfer to a susceptible line of barley. In addition, the product elicits a hypersensitive reaction with transiently expressed in *Nicotiana benthamiana*. The authors also present an extensive analysis of related coding sequences in related species. Comparisons are made to so-called executor R genes of rice and pepper, which are targeted and induced by effectors that act as transcription factors. Speculation is provided that the cognate effector may function in a similar fashion.

The new class is an exciting find and presents an occurrence of a R gene directed to a fungus that requires induction of expression. The EMS mutagenesis and gene transfer provide strong evidence that the ORF2 is the gene. The results are of considerable significance.

The main criticism/suggestions of the work is the length and lack of basic promoter analysis given the proposed requirement for pathogen-specific gene induction. More to the point, three items would enhance the manuscript.

Response: We thank the referee for their constructive comments. We have considered all comments as follows.

Comment: The manuscript could be shortened considerably. The discussion is very long given little analysis of the mode of gene induction. The discussion of TALE-mediated expression presents too much detail, for example, given the speculative nature of the hypothesis.

Response: We have shortened the discussion section by deleting the paragraph about the NCED gene in page 14, lines 447 - 456, and lines 458 - 461. We also shortened the last paragraph of the discussion section in page 15, lines 497 - 507 of the revised version.

Comment: The authors should measure the expression of *Rph3* in the transgenic progeny.

Response: We have performed the expression analysis of *Rph3* in the transgenic progenies. The result is added in page 7, lines 207 - 212 of the revised version and data is shown in Figure S19.

Comment: An analysis of the promoter region and comparisons of the promoter region in the various germplasm and lines should have been presented with the protein analysis since

much discussion in the MS was devoted to the requirement for gene induction. Many aspects for promoter function can be obtained by comparative and consensus analyses.

Response: This comment is much appreciated, and we fully agree with the importance of promoter analysis for this inducible gene. However, comparative sequence analysis will remain descriptive until functional tests with overlapping series of promoter lesions are performed, which is beyond the scope of the current manuscript.

Comment: line 98 typo mino to amino

Response: the word "mino" has been changed into "amino" in page 4, line 100 of the revised version.

Comment: In discussion, ABA has documented effects both for enhanced susceptibility upon expression or resistance upon suppression. (As noted above, this aspect of discussion can be removed as very speculative. Homology to portions to NCED, which is related to larger set of proteins does not readily translated to increased ABA.)

Response: This paragraph has been deleted in page 14, lines 447 - 456 of the revised version according to this comment.

Comment: The reference list is very large, and, again, could be reduced by reduction of the discussion.

Response: By deleting several sentences in the discussion section, more than 10 references have been removed from this section.

Comment: It is somewhat unfortunate the name "executor" is used for two reasons. One, the "executor" comparison is still speculative at this point in regard to details, and, two, the term probably should be "executioner". But be that as it may, the term has been adopted. Cannot the title just claim a novel R gene.

Response: We fully agree with this comment. However, the word executor has been used in literature, and we kept it as it is for consistency purpose.

Reviewer #3 (Remarks to the Author):

Comment: The manuscript entitled “The barley leaf rust resistance gene Rph3 encodes a putative executor protein” by Dinh et al describes molecular cloning and characterization of a putative new executor R gene Rph3 from barley. This is a very interesting work because the R gene Rph3 confers resistance to barley leaf rust caused by fungal pathogen *Puccinia hordei*, not the bacterial pathogen *Xanthomonas* spp. that are specifically associated with the so-called executor R genes, previously.

The Rph3 (Reaction to *Puccinia hordei* 3) locus was first discovered in barley landrace ‘Estate’ using classical genetics in 1967. Although pathotypes with virulence for Rph3 have been detected in all barley growing areas, Rph3 remains a valuable source of resistance for barley breeding. The authors of this manuscript isolated the Rph3 gene by map-based cloning, confirmed by mutational analysis and transgenic complementation; the conclusion on Rph3 cloning was strongly supported by abundant data.

What’s interesting is that the Rph3 gene encodes a unique transmembrane resistance protein differing from all known plant disease resistance proteins at the amino acid sequence level. The authors clearly demonstrated that the Rph3 gene was expressed only in interactions with Rph3-avirulent isolates of *Puccinia hordei*, a phenomenon observed for the transcription activator-like effector-dependent executor R genes conferring resistance to *Xanthomonas*. The Rph3 gene also possesses some other features of the known executor R genes isolated from pepper and rice, such as encoding a protein with transmembrane-helix domains, expression induces cell death response in *Nicotiana benthamiana* as well as in host plants. Based on these observations, the authors concluded that ‘executor’ genes exist in the Triticeae tribe, and executor genes (e.g., Rph3) could confer resistance against fungal pathogen.

This work is important because it put new insights to better understanding the scope and resistance mechanism of executor R genes, as well as facilitating the application of the valuable R gene Rph3 in barley breeding for leaf rust resistance.

Response: Thank you for your description and constructive comments. We have considered all comments as follows.

Comment: The major limitation of this work is the lack of information about the cognate effector (AvrRph3) in *P. hordei*, which should specifically activate expression of Rph3. This effector is a critical factor in determining Rph3 as an executor gene.

Response: We fully agree. Therefore, we have had a sentence: “Further work is required to demonstrate this, in particular, the isolation of *AvrRph3*. ” in page 12, line 390 - 391. We also add another sentence: “Further studies are required to test these hypotheses.” in page 15, line 503.

Comment: If possible, provide the evidence of mis-spliced RNAs in the M466 mutant.

Response: A sentence has been added in page 6, lines 170 - 171 of the revised version: “The mutant line M466 did not survive, and the changes in the protein structure of the RPH3 protein could not be examined.”

Comment: The result section “Transcription dynamics of Rph3-mediated resistance at two days post-inoculation” does not provide explicit genes associated with pathways or mechanism of Rph3-mediated resistance. It could be moved into discussion.

Response: A sentence has been added in page 10, lines 312 - 313 of the revised version: “However, the actual genes associated with pathways, or the mechanism of *Rph3*-mediated resistance could not be determined due to the large number of DEG.”

Comment: Line 117: “Figure 2c)” should be Figure 1c.

Response: “2c” has been changed into “1c” in page 5, line 116 of the revised version.

Comment: Line 122: Better to change the “higher response” as “higher susceptibility”.

Response: The text has been changed into “... an intermediate response to those of BW746 (resistant) and Bowman (susceptible) ...” in page 5, lines 130 - 131 of the revised version.

Comment: Line 621-623: The 7,096 bp genomic DNA segment for Rph3 construct was synthesized? Please provide more information about the synthesis.

Response: More details about the synthesis of the DNA segment for *Rph3* construct have been added in the SI Materials and Methods section, page 20, lines 667 - 671 of the revised version.

Comment: Line 1116/Fig. SI1: It seems Bowman and BW746 were wrongly positioned?

Response: We have replaced Fig. SI1 with a better quality and adjusted its legends to clarify the purpose in page 35 - 36, lines 1196 - 1206 of the revised version.

Comment: Line 1216-1218/Fig.SI14: The mentioned black/green/black arrows can't be found in the figure?

Response: The gene models with their introns/exons in black/green have been added to the figure named as Fig. SI17 in the revised version.

REVIEWERS' COMMENTS

Reviewer #1 (Remarks to the Author):

The authors have addressed all my concerns on the original version of the manuscript.

I have no further comments.

Reviewer #2 (Remarks to the Author):

The manuscript reports the identification of a new R gene in barley. The gene is novel with regards to structure and expression. The identification of the R gene is a significant result, and the authors provided some updates, for example, the transcription analysis of the transgenics. The work indicates that the gene (Rph3) is only expressed upon infection with an avirulent fungal isolate, containing, hypothetically, a cognate Avr gene/effector (AvrRph3). The authors hypothesize that gene induction of Rph3 is mediated by AvrRph3 in a manner similar to the TALE/executor R gene pairs of the bacterial pathogen *Xanthomonas*. This is possible, but the data to support it is non-existent other than the expression data. This reviewer does not like the title, although it depends on one's definition of an executor R gene. Does it just have to be inducible? Or, does it have to be induced by a transcription factor from the pathogen? I will leave that to others. Cannot the title simple be the isolation of a new R gene family that is expressed only in an incompatible response? The other model is that the difference is due to a cascade of expression events resulting from recognition by another component, possible an NBS-LRR or RLK, which in themselves, do not trigger resistance. R are not strictly NBS-LRR or RLK genes but simply the genetic difference between compatible and incompatible isolates of a pathogen. While commonly NBS-LRRs or RLKs, not exclusively so.

Other items:

1. Define p+ and P-
2. Remains quite long despite claims
3. What does "aligned" with pathogen mediated expression in the transgenics? What about transgenic E26-06 R with MOC400 marker or expression of Rph3.
4. Again, this review still does not see why comparisons of the promoter region is not included in the analyses. The authors indicate it would not lead to any insight. It provides just as much insight as protein structure does. So much attention is given to the expression and not requirements of the structure, yet the structure is given all the analysis....

5. The RNAseq adds nothing to the paper except length. It also lacks the virulent isolate interaction set as noted by other reviewer. Should be removed.

Reviewer #3 (Remarks to the Author):

Based on the revised manuscript and the point-by-point response from the authors, all my concerns have been well addressed. Therefore, I recommend the acceptance of the revised manuscript for publication in this prestigious journal.

Reviewer #1 (Remarks to the Author):

Comment: The authors have addressed all my concerns on the original version of the manuscript. I have no further comments.

Response: We thank Reviewer 1 for the constructive comments that have significantly improved our manuscript. We are pleased that all the concerns have been fully addressed.

Reviewer #2 (Remarks to the Author):

Comment: The manuscript reports the identification of a new R gene in barley. The gene is novel with regards to structure and expression. The identification of the R gene is a significant result, and the authors provided some updates, for example, the transcription analysis of the transgenics. The work indicates that the gene (Rph3) is only expressed upon infection with an avirulent fungal isolate, containing, hypothetically, a cognate Avr gene/effector (AvrRph3). The authors hypothesize that gene induction of Rph3 is mediated by AvrRph3 in a manner similar to the TALE/executor R gene pairs of the bacterial pathogen *Xanthomonas*. This is possible, but the data to support it is non-existent other than the expression data. This reviewer does not like the title, although it depends on one's definition of an executor R gene. Does it just have to be inducible? Or, does it have to be induced by a transcription factor from the pathogen? I will leave that to others. Cannot the title simple be the isolation of a new R gene family that is expressed only in an incompatible response? The other model is that the difference is due to a cascade of expression events resulting from recognition by another component, possible an NBS-LRR or RLK, which in themselves, do not trigger resistance. R are not strictly NBS-LRR or RLK genes but simply the genetic difference between compatible and incompatible isolates of a pathogen. While commonly NBS-LRRs or RLKs, not exclusively so.

Response: We thank Reviewer 2 for constructive comments and have changed the title accordingly. All the conclusive statements on Rph3 as an executor has been removed throughout the manuscript in lines 1-2, 46-48, 50, 101-102, 274-275, 359-361, 430, 443, and 479 of the revised version.

Comment: Define p+ and P-

Response: The symbols P+/P- were defined in lines 524-525 of the current version.

Comment: Remains quite long despite claims

Response: We have moved lines 290-297 of RNAseq results to Supplementary Figure 16 legend.

Comment: What does "aligned" with pathogen mediated expression in the transgenics? What about transgenic E26-06 R with MOC400 marker or expression of Rph3.

Response: The explanation for the phenotype of transgenic plant E25-06 has been added to the revised version's legend of the Supplementary Figure 9.

Comment: Again, this review still does not see why comparisons of the promoter region is not included in the analyses. The authors indicate it would not lead to any insight. It provides just as much insight as protein structure does. So much attention is given to the expression and not requirements of the structure, yet the structure is given all the analysis....

Response: The promoter analysis has been performed, the text has been added to page 9-10, lines 312-316, the Supplementary Figure 18 has been added to the Supplementary Information, and the method has been added to page 21, lines 709-714 of the revised version.

Comment: The RNAseq adds nothing to the paper except length. It also lacks the virulent isolate interaction set as noted by other reviewers. Should be removed.

Response: We have moved lines 290-297 of RNAseq results to Supplementary Figure 16 legend.

Reviewer #3 (Remarks to the Author):

Comment: Based on the revised manuscript and the point-by-point response from the authors, all my concerns have been well addressed. Therefore, I recommend the acceptance of the revised manuscript for publication in this prestigious journal.

Response: We thank Reviewer 3 for valuable comments and time. We are pleased that all the comments have been well addressed.